# Maternal exposure to SSRIs or SNRIs and the risk of congenital abnormalities in offspring: A systematic review and meta-analysis

Weiyi Huang[1], Robin L. Page[2]*, Theresa Morris[3], Susan Ayres[4], Alva
O. Ferdinand[5], Samiran Sinha[6]*

1 Department of Epidemiology and Biostatistics, Texas A&M University, College Station, TX, United States of America, 2 School of Nursing, Texas A&M University, College Station, TX, United States of America, 3 Department of Sociology, Texas A&M University, College Station, TX, United States of America, 4 School of Law, Texas A&M University, Fort Worth, TX, United States of America, 5 Southwest Rural Health Research Center, Texas A&M University, College Station, TX, United States of America, 6 Department of Statistics, Texas A&M University, College Station, TX, United States of America

* sinha@stat.tamu.edu (SS); rpage@tamu.edu (RLP)

**Data Availability Statement:** All relevant data are within the paper and its Supporting information files.

## Abstract

### Background

The association of maternal exposure to selective serotonin reuptake inhibitors (SSRIs) or serotonin and norepinephrine reuptake inhibitors (SNRIs) with the risk of system-specific congenital malformations in offspring remains unclear. We conducted a meta-analysis to examine this association and the risk difference between these two types of inhibitors.

### Methods

A literature search was performed from January 2000 to May 2023 using PubMed and Web of Science databases. Cohort and case-control studies that assess the association of maternal exposure to SSRIs or SNRIs with the risk of congenital abnormalities were eligible for the study.

### Results

Twenty-one cohort studies and seven case-control studies were included in the meta-analysis. Compared to non-exposure, maternal exposure to SNRIs is associated with a higher risk of congenital cardiovascular abnormalities (pooled OR: 1.64 with 95% CI: 1.36, 1.97), anomalies of the kidney and urinary tract (pooled OR: 1.63 with 95% CI: 1.21, 2.20), malformations of nervous system (pooled OR: 2.28 with 95% CI: 1.50, 3.45), anomalies of digestive system (pooled OR: 2.05 with 95% CI: 1.60, 2.64) and abdominal birth defects (pooled OR: 2.91 with 95%CI: 1.98, 4.28), while maternal exposure to SSRIs is associated with a higher risk of congenital cardiovascular abnormalities (pooled OR: 1.25 with 95%CI: 1.20, 1.30), anomalies of the kidney and urinary tract (pooled OR: 1.14 with 95%CI: 1.02, 1.27), anomalies of digestive system (pooled OR: 1.11 with 95%CI: 1.01, 1.21), abdominal birth defects (pooled OR: 1.33 with 95%CI: 1.16, 1.53) and musculoskeletal malformations (pooled OR: 1.44 with 95%CI: 1.32, 1.56).

**Funding:** This work was funded by an internal grant from Texas A&M University Division of Research and was awarded to the following authors: RP, TM, SA, AF, SS. The Grant number is 290414-00001. The URL for the funder is https://vpr.tamu.edu/find-funding/. The funders had no role in study design, data collection and analysis, decision to publish, or preparation of the manuscript.

**Competing interests:** The authors have declared that no competing interests exist.

## Conclusions

SSRIs and SNRIs have various teratogenic risks. Clinicians must consider risk-benefit ratios and patient history when prescribing medicines.

## Introduction

Across the nation, birth defects are the leading cause of infant death, accounting for 20 percent of all infant deaths [1], so preventing birth defects is an important social goal. Yet, the etiology of birth defects is complex, which means that there are many pathways for improvement. One important focus is on the effect of SSRIs and SNRIs, used to treat depression, when taken during pregnancy. Depression is a common mood disorder. Since the start of the COVID-19 pandemic, more people, especially women, have suffered from depression and sought therapy for it [2]. In the United States, depression affects 10% to 20% of women during pregnancy, the postpartum period, or both [3], with higher rates reported among pregnant individuals during the COVID-19 pandemic [4]. Antidepressants are evidence-based treatments for depression during pregnancy. However, antidepressants' adverse associations with fetal and infant health, including the possibility of congenital disabilities following exposure during pregnancy, remains a source of concern [5]. To provide evidence-based guidance for therapeutic management during pregnancy, it is necessary to investigate the relationships between antidepressant use during pregnancy and congenital abnormalities.

Selective serotonin reuptake inhibitors (SSRIs) are the most common antidepressants prescribed during pregnancy, followed by serotonin and norepinephrine reuptake inhibitors (SNRIs) [6]. The chemical serotonin is responsible for regulating mood, sleep, bone health, wound healing, sexual drive and many more physiological and psychological activities. SSRIs treat depression by blocking the reuptake of serotonin by neurons and increasing levels of serotonin in brain, which makes more serotonin available to improve transmission of messages between neurons [7]. The chemical norepinephrine is responsible for maintaining blood pressure, increasing attention and regulating sleep-wake cycle, mood and memory. SNRIs help relieve depression by blocking the reuptake of both serotonin and norepinephrine in brain and affecting neurotransmitters used to communicate between brain cells, which ultimately affects changes in brain chemistry and communication in brain nerve cell circuitry known to regulate mood [8].

Serotonin is not only responsible for psychological activities and processes. Studies also suggest that serotonin plays an important role in fertility and normal embryonic development as different serotonin receptor subtypes can be found in human oocytes and granulosa cells that leads to the production of progesterone and estrogen [9, 10]. Norepinephrine plays an important role in arousal, attention, cognitive function and the body's reaction to an emergency situation. Norepinephrine also plays a critical role in glycogenolysis and gluconeogenesis (while reducing glucose clearance) and inducing ketogenesis and lipolysis [11, 12]. These all may have detrimental effects on embryonic development. Over the past few decades, there has been a rise in women's use of antidepressants during pregnancy. The most frequently prescribed antidepressants during pregnancy are selective serotonin reuptake inhibitors (SSRIs), with an estimated global prevalence of 3% and a North American prevalence of 5.5% [6]. For serotonin and norepinephrine reuptake inhibitors (SNRIs), the second most frequently prescribed antidepressants during pregnancy, the international prevalence estimate is 0.73% [6]. Therefore, studies estimating the risk of these antidepressants to the fetus are critical for providing evidence-based clinical guidelines for health care practitioners.

In the overarching goal of understanding the role of SSRI and SNRI on congenital development, many studies have analyzed the association of maternal exposure to SSRIs with adverse neonatal or pregnancy outcomes [13, 14]. In contrast, studies on the association of maternal exposusre to SNRIs with neonatal or pregnancy outcomes are limited [5] or do not separate the risk assessment of SNRIs from SSRIs [15]. Previous studies have indicated the differences in the risk of adverse clinical outcomes associated with SNRIs compared to SSRIs for cerebrovascular, cardiovascular, and mortality events [16] and the risk of fractures among 50 years or older adults [17]. But such a comparison has not been reported in congenital disabilities (commonly known as birth defects). Gao et al. [18] conducted a meta-analysis on the association of maternal exposure to SSRIs with the risks of 29 categories of congenital malformations. Still, there is no such meta-analysis on SNRIs. De Vries et al. [5] conducted a meta-analysis on the association of maternal antidepression exposure (SSRIs or SNRIs) with the risks of congenital heart defects without studying the differential effects of SSRIs and SNRIs. Ours is the first meta-analysis to examine the association of maternal exposure to SNRIs with system-specific malformations in offspring and the first to explore the difference between SNRIs and SSRIs in terms of the risk of congenital abnormalities in offspring.

## Methods

### Literature search, inclusion criteria and quality assessment

Two independent authors conducted a literature search examining articles published between January 2000 to May 2023 using the PubMed and the Web of Science databases. Medical Subject Heading (MeSH) terms, Key words and Boolean operators were used to form the search strategy S1 Appendix. Cohort studies and case-control studies that were published in English were included. Case reports, reviews, conference abstracts and basic science research were excluded. Eligible studies included (a) SSRIs or SNRIs as the exposure of interest, (b) congenital malformation as the outcome of interest, and (c) provided raw counts to calculate the crude odds ratio (cOR) and confidence intervals (CIs). Nine system-specific congenital malformations were analyzed separately, including cardiovascular abnormalities; kidney and urinary tract anomalies; nervous system malformations; digestive system anomalies; abdominal birth defects; musculoskeletal malformations; eye, ear, face and neck malformations; genital organs malformations; respiratory system malformations. Additionally, overall malformations that included any of the nine malformations were also analyzed. Newcastle-Ottawa Scale [19] was used to assess study quality. A study was judged on three broad perspectives: the selection of the study groups; the comparability of the groups; and the ascertainment of either the exposure or outcome of interest for case-control or cohort studies, respectively. A study was categorized as high quality if scored $\geq 8$ points, fair quality if scored 6–7 points, and poor quality if scored $<6$ points.

### Statistical analysis

STATA/MP 17.0 (Stata Corp. 2021, LLC) was used for data analyses. Pooled risk estimates with 95% confidence intervals (CIs) were obtained using a fixed effect model. When significant heterogeneity was present, we used a random effect model to estimate [20]. We used the $I^2$ statistic to measure the heterogeneity among studies, and results were considered as heterogeneous for $I^2 >50\%$ and the corresponding P $<0.05$ [21]. The statistic $I^2$ can be seen as a relative measure of heterogeneity to the total variability. We also provided the $\tau^2$ estimate which is an absolute measure of between-study variability. The funnel plot and Egger's regression tests assessed small-study effects, such as publication bias [22]. The funnel plot is a scatter plot of log (odds ratio) against the standard error for different studies. In the ideal case, the

points should be equally distributed around the logarithm of the pooled odds ratio line. In contrast, an asymmetrical distribution of the points around that line indicates a possible publication bias. Besides this visual procedure, Egger's test objectively identifies the publication bias. The trim-and-fill analyses were used to adjust for the potential publication bias [23] when the bias may exist, and pooled risk estimates with 95% confidence intervals (CIs) were recalculated after the trim-and-fill analyses.

## Results

### Characteristics of included studies

The literature search identified 2046 records from the two databases. After we screened the titles and abstracts based on the inclusion criteria, 2013 records were excluded for duplicates, conference abstract, case report, bench research, non-relevant to the study topic, etc. After the full text review of 33 studies to assess for eligibility, 5 studies, which did not specify categories of congenital malformations, were further excluded. In the end, we included 7 case-control studies and 21 cohort studies in the final analysis Fig 1. Of the 28, 14 studies were conducted in European countries, 9 studies were conducted in the US, 2 studies were conducted in Japan,

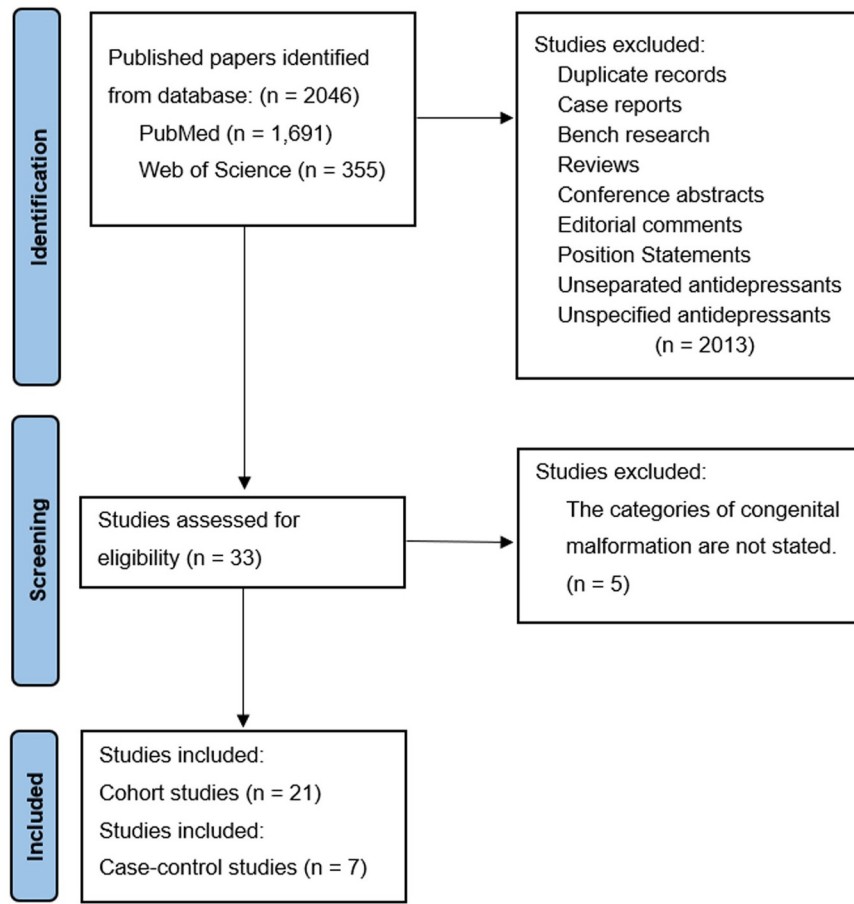

**Fig 1. PRISMA flow diagram of evidence-based literature search and selection for SSRIs/SNRIs and congenital abnormalities.**

**Table 1. Characteristics of studies included in the meta-analysis.**

| Study (Year) | Study Type | Location | Sample size or Number of case/control | Duration | Exposure | Outcome (Malformation subtype) |
|---|---|---|---|---|---|---|
| Ankarfeldt (2021) [24] | Cohort | Denmark, Sweden | 2,076,164 | 2004–2016 | Duloxetine | Heart; Digestive system; Ear, face and neck; Eye; Genitals; Nervous system; Limb; Oro-facial clefts; Respiratory system; Urinary tract; Abdominal wall |
| Kolding (2021) [25] | Cohort | Denmark | 364,012 | 2007–2014 | Venlafaxine, SSRI | Cardiac malformations |
| Anderson (2020) [26] | Case-Control | US | 30,630/11,478 | 1997–2011 | Fluoxetine, Sertraline, Citalopram, Venlafaxine | Heart defect; Neural tube defect; Hydrocephaly; Dandy-Walker malformation; Cataracts; Glaucoma/anterior chamber defects; Anotia/microtia; Oral cleft; Esophageal atresia; Intestinal atresia/stenosis; Duodenal atresia/stenosis; Anorectal atresia/stenosis; Hypospadias; Limb deficiencies; Craniosynostosis; Diaphragmatic hernia; Omphalocele; Gastroschisis; Biliary atresia; Bilateral renal agenesis or hypoplasia; Choanal atresia |
| Huybrechts (2020) [27] | Cohort | US | 1,287,359 | 2004–2013 | Duloxetine | Cardiovascular malformations |
| Yamamoto-Sasaki (2020) [28] | Cohort | Japan | 53,638 | 2005–2014 | SSRI, SNRI | Eye; Ear, face and neck; Heart; Circulatory system; Orofacial clefts; Musculoskeletal system; Integument; Genital organs |
| Werler (2018) [29] | Case-Control | US | 1,261/10,682 | 1997–2011 | Venlafaxine | Gastroschisis |
| Nielsen (2017) [30] | Cohort | Denmark | 1,256,317 | 1996–2016 | SSRI | Hirschsprung's disease |
| Nishigori (2017) [31] | Cohort | Japan | 95,994 | 2011–2014 | SSRI | Upper limb; Abdominal wall; Urogenital system |
| Petersen (2016) [32] | Cohort | UK | 209,135 | 1990–2011 | SSRI | Heart anomalies |
| Jordan (2016) [33] | Cohort | Norway, Denmark, Wales | 519,117 | 2000–2010 | SSRI | Neural tube defects; Heart defects; Abdominal wall defects; Talipes equinovarus; Hypospadias; Ano-rectal atresia and stenosis; Renal dysplasia; Limb reduction; Craniosynostosis |
| Bérard (2015) [34] | Cohort | Canada | 18,493 | 1998–2010 | Sertraline, SSRI | Nervous system; Eye, ear, face and neck; Circulatory system; Respiratory system; Digestive system; Genital organs; Urinary system; Musculoskeletal system; Cardiac system; Ventricular/atrial septal defect; Omphalocele; Craniosynostosis; Cleft palate |
| Furu (2015) [35] | Cohort | Denmark, Finland, Iceland, Norway, Sweden | 2,303,647 | 1996–2010 | Fluoxetine, Citalopram, Sertraline, Venlafaxine | Cardiac defects; Anal atresia; Hypospadias; Clubfoot; Limb reduction; Craniosynostosis |
| Huybrechts (2014) [36] | Cohort | US | 949,504 | 2000–2007 | Sertraline, Fluoxetine, SNRI | Cardiac defects |
| Ban (2014) [37] | Cohort | UK | 349,127 | 1990–2009 | SSRI | Heart; Limb; Genital system; Urinary system; Orofacial cleft; Nervous system; Musculoskeletal; Digestive system; Eye; Respiratory system; Abdominal wall |
| Knudsen (2014) [38] | Cohort | Denmark | 72,280 | 1995–2008 | SSRI | Heart defects |
| Yazdy (2014) [39] | Case-control | US | 622/2002 | 2006–2011 | Fluoxetine, Sertraline, Citalopram | Clubfoot |
| Vasilakis-Scaramozza (2013) [40] | Cohort | UK | 8,442 | 1991–2002 | SSRI | Central nervous system; Ear, eye, face and neck; Cardiovascular; Genital anomalies; Urinary anomalies |
| Margulis (2013) [41] | Cohort | UK | 12,037 | 1996–2010 | SSRI | Cardiac malformation |

*(Continued)*

**Table 1.** (Continued)

| Study (Year) | Study Type | Location | Sample size or Number of case/control | Duration | Exposure | Outcome (Malformation subtype) |
|---|---|---|---|---|---|---|
| Lind (2013) [42] | Casee-control | US | 1537/4314 | 1997–2007 | Venlafaxine | Hypospadias |
| Polen (2013) [43] | Case-control | US | 19,043/8002 | 1997–2007 | Venlafaxine | Spina bifida; Anotia or Microtia; Construal heart defects; Septal heart defects; Perimembranous ventricular septal defects; Atrial septal defects; Ventricular septal defects; Right ventricular outflow tract obstruction defects; Pulmonary valve stenosis; Left ventricular outflow tract obstruction defects; Coarctation of the aorta; Cleft lip with or without cleft palate; Cleft palate alone; Hypospadias; Limb reduction defects; Craniosynostosis; Diaphragmatic hernia; Gastroschisis |
| Nordeng (2012) [44] | Cohort | Norway | 63,395 | 2000–2006 | Fluoxetine, Sertraline, Citalopram | Cardiovascular malformation |
| Malm (2011) [45] | Cohort | Finland | 635,583 | 1996–2006 | Citalopram, Fluoxetine, Sertraline | Cardiovascular malformation |
| Kornum (2010) [46] | Cohort | Denmark | 216,042 | 1991–2007 | SSRI | Cardiac malformation |
| Pedersen (2009) [47] | Cohort | Denmark | 493,113 | 1996–2003 | Fluoxetine, Citalopram | Eye; Cardiac malformations; Cleft lip with or without cleft palate; Cleft palate alone; Gastrointestinal; Craniosynostosis |
| Merlob (2009) [48] | Cohort | Israel | 67,871 | 2000–2007 | SSRI | Cardiac malformations |
| Diav-Citrin (2008) [49] | Cohort | Israel, Italy, Germany | 1,612 | 1994–2002 | Fluoxetine | Cardiovascular anomalies; Genitourinary anomalies |
| Alwan (2007) [50] | Case-control | US | 9622/4092 | 1997–2002 | SSRI | Anencephaly; Spina bifida; Conotruncal heart defects; Septal heart defects; Right ventricular outflow tract obstruction defects; Left ventricular outflow tract obstruction defects; Cleft lip with or without cleft palate; Cleft palate alone; Esophageal atresia; Anorectal atresia; Hypospadias; Limb deficiencies; Craniosynostosis; Omphalocele; Diaphragmatic hernia; Gastroschisis |
| Louik (2007) [51] | Case-control | US | 9849/5860 | 1993–2004 | Fluoxetine, Sertraline, Citalopram | Craniosynostosis; Omphalocele; Cardiac defects; Cleft lip with or without cleft palate; Pyloric stenosis; Renal-collecting-system defects; Hypospadias; Clubfoot; Cleft palate alone; Undescended testis; Neural-tube defects; Anal atresia; Diaphragmatic hernia; Limb reduction defects |

1 study was conducted in Canada, 1 study was conducted in Israel, and 1 study was conducted in European countries and Israel Table 1. All the 28 included studies were assessed as high-quality studies except for one fair quality study. These studies reported the associations of maternal exposure to SSRI or SNRI with specific congenital malformations in offspring. All the congenital malformations cases were clinically diagnosed and classified based on ICD-9 codes to identify the congenital malformations by organ system. Most studies were judged as high-quality studies with high Newcastle-Ottawa scores, which indicated relatively low risk of bias within these studies, see S1 and S2 Tables.

## Risk of congenital cardiovascular abnormalities

Twenty studies assessed maternal exposure to SSRIs and the risk of cardiovascular anomalies in offspring, and nine studies assessed maternal exposure to SNRIs and the risk of

cardiovascular malformations in offspring. These studies reported common congenital cardio-vascular anomalies: tetralogy of Fallot, transposition of the great arteries, ventricular/atrial septal defect, left/right ventricular outflow tract obstruction, etc.

Compared to the offspring of women not exposed to SSRIs or SNRIs, the offspring of women exposed to SSRIs or SNRIs were at higher risk of congenital cardiovascular abnormalities. The pooled OR was 1.25 (95% CI: 1.20, 1.30) for those women exposed to SSRIs during pregnancy, with the reported ORs ranging from 0.52 (95% CI: 0.19, 1.44) to 4.81 (95% CI: 1.73, 13.37) Fig 2. The pooled OR was 1.72 (95% CI: 1.53, 1.93) for those women exposed to SNRIs during pregnancy, with the reported ORs ranging from 1.23 (95% CI: 0.90, 1.68) to 5.25 (95% CI: 0.72, 38.48) Fig 3. No significant heterogeneity was detected in either group at the 5% level of significance ($I^2$ = 0.0%, P = 0.523; $I^2$ = 35.1%, P = 0.076). For those women exposed to SSRIs, no publication bias was detected as indicated by the funnel plot and Egger's test (Coefficient = -0.03, 95% CI: -0.52, 0.47; P = 0.913) Fig 4. However, for those women exposed to

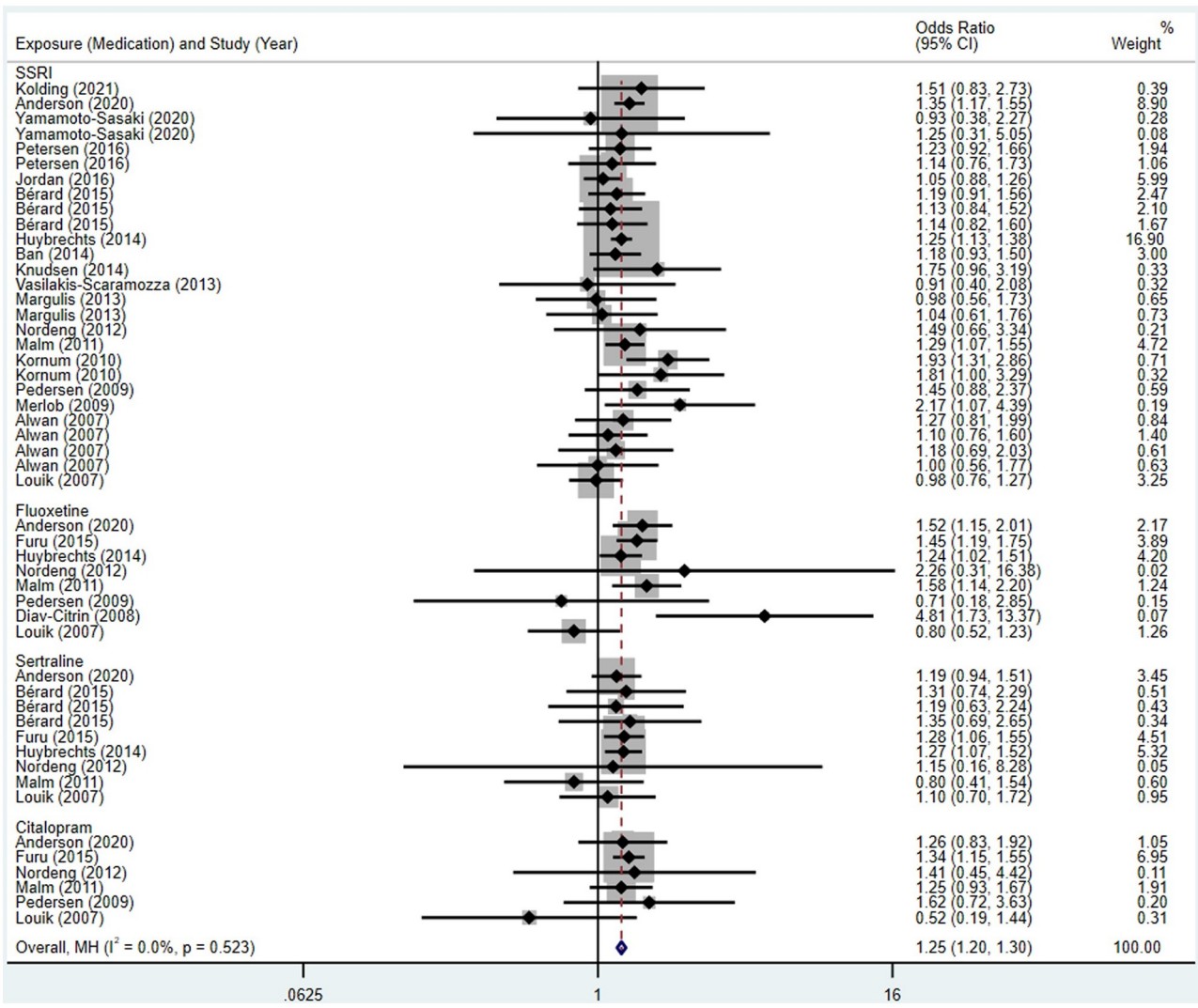

**Fig 2. Forest plot of the association between maternal SSRIs exposure and risk of congenital cardiovascular abnormalities in offspring.**

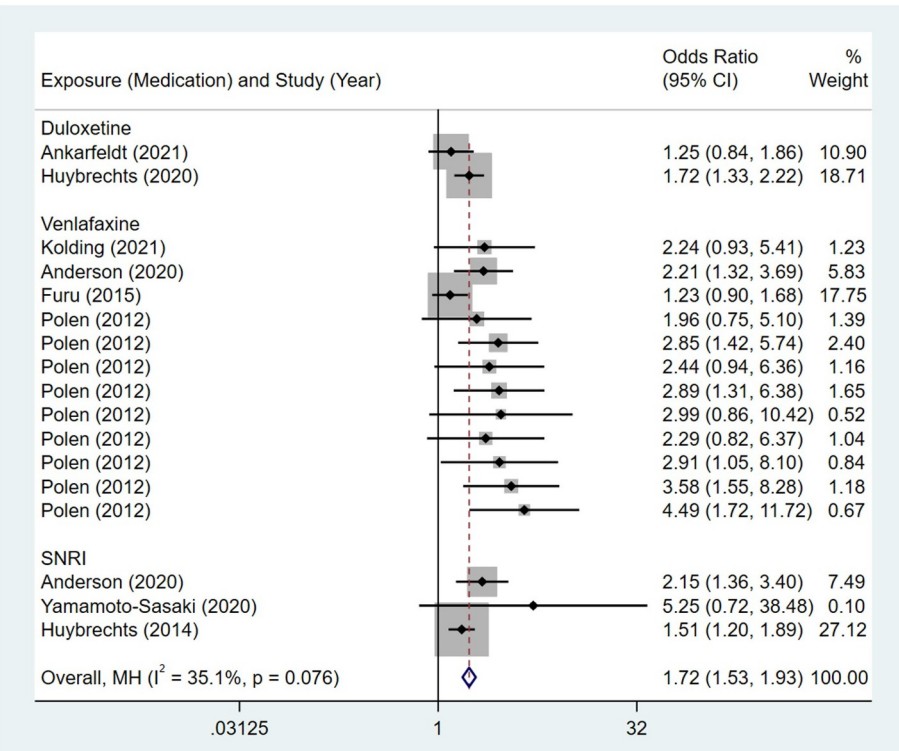

**Fig 3. Forest plot of the association between maternal SNRIs exposure and risk of congenital cardiovascular abnormalities in offspring.**

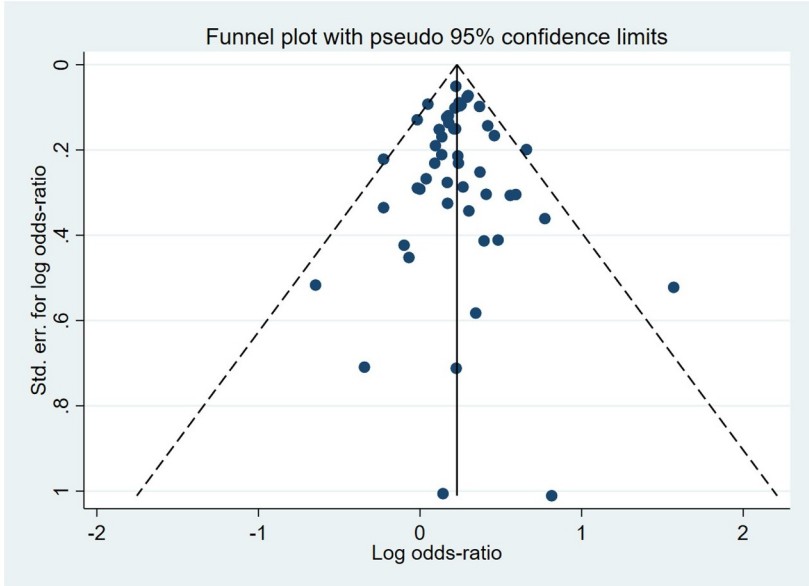

**Fig 4. Funnel plot of the association between maternal SSRIs exposure and risk of congenital cardiovascular abnormalities in offspring.**

SNRIs, there was a publication bias as indicated by the funnel plot and Egger's test (Coefficient = 1.73, 95% CI: 0.93, 2.52; P <0.001). When we used the trim and fill method to adjust for the potential publication bias, the recalculated pooled odds ratio of the association of maternal SNRIs exposure with cardiovascular malformations in offspring was 1.64 (95% CI: 1.36, 1.97) Fig 5.

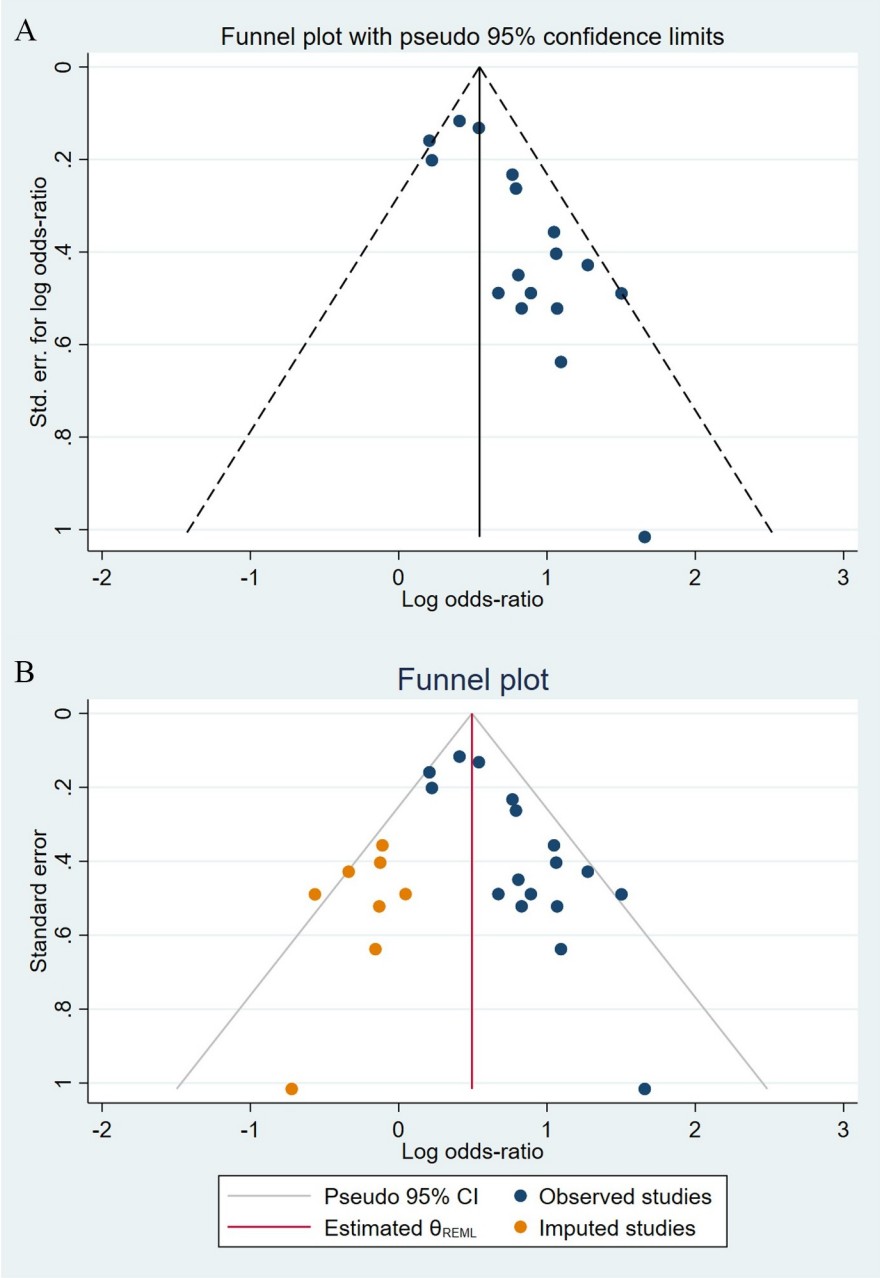

**Fig 5. Funnel plot of the association between maternal SNRIs exposure and risk of congenital cardiovascular abnormalities in offspring.** A: Funnel plot before trim-and-fill methods. B: Funnel plot after trim-and-fill methods.

### Risk of congenital anomalies of the kidney and urinary tract

Eight studies assessed maternal exposure to SSRIs and the risk of congenital anomalies of the kidney and urinary tract in offspring, and five studies assessed maternal exposure to SNRIs and the risk of congenital anomalies of the kidney and urinary tract in offspring. These studies reported birth defects involving hypospadias, renal agenesis/dysplasia and renal-collecting-system defects.

Compared to the offspring of women not exposed to SSRIs or SNRIs, the offspring of women exposed to SSRIs or SNRIs were at higher risk of kidney and urinary tract anomalies. The pooled OR was 1.14 (95% CI: 1.02, 1.27) for those women exposed to SSRIs during pregnancy, with the reported ORs ranging from 0.37 (95% CI: 0.17, 0.80) to 3.16 (95% CI: 1.05, 9.56) Fig 6. The pooled OR was 1.72 (95% CI: 1.25, 2.35) for those women exposed to SNRIs during pregnancy, with the reported ORs ranging from 1.27 (95% CI: 0.60, 2.67) to 2.55 (95% CI: 1.03, 6.34) Fig 7. No significant heterogeneity was detected in either group ($I^2$ = 16.8%, P = 0.233; $I^2$ = 0.0%, P = 0.851). For those women exposed to SSRIs, no publication bias was detected as indicated by the funnel plot and Egger's test (Coefficient = -0.65, 95% CI: -1.67, 0.36; P = 0.194) Fig 8. However, for those women exposed to SNRIs, there was a publication bias as indicated by the funnel plot and Egger's test (Coefficient = 5.03, 95% CI: 0.94, 9.13; P = 0.027). When we used the trim and fill method to adjust for the potential publication bias, the recalculated pooled odds ratio of the association of maternal SNRIs exposure with the kidney and urinary tract anomalies in offspring was 1.63 (95% CI: 1.21, 2.20) Fig 9.

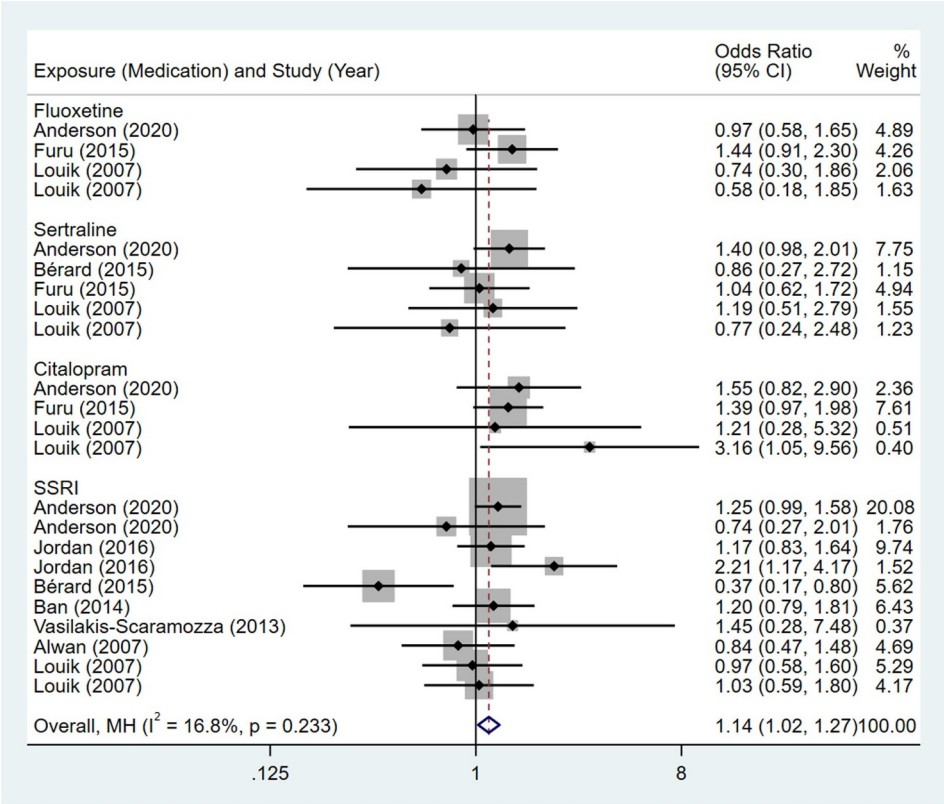

**Fig 6. Forest plot of the association between maternal SSRIs exposure and risk of congenital kidney and urinary tract abnormalities in offspring.**

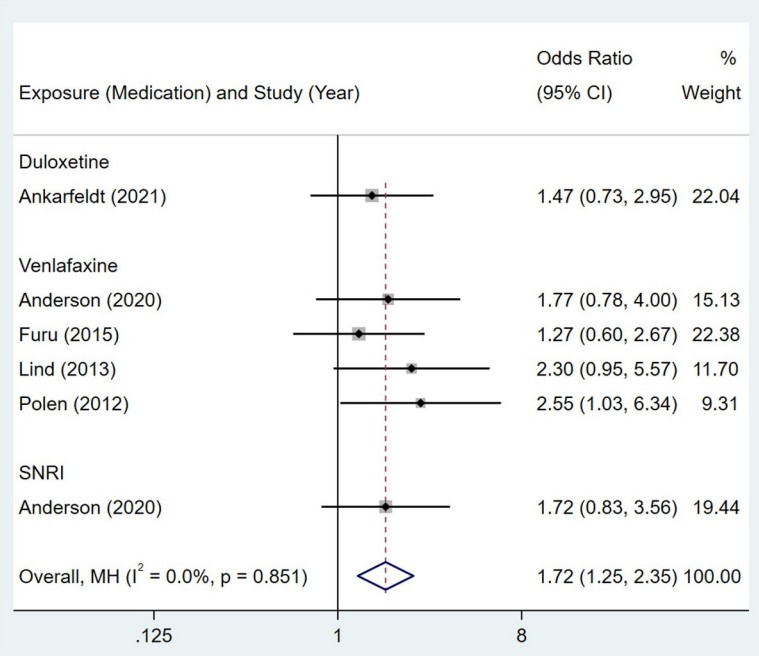

**Fig 7. Forest plot of the association between maternal SNRIs exposure and risk of congenital kidney and urinary tract abnormalities in offspring.**

## Risk of congenital malformations of the nervous system

Eight studies assessed maternal exposure to SSRIs and the risk of congenital anomalies of the nervous system in offspring, and three studies assessed maternal exposure to SNRIs and the risk of congenital anomalies of the nervous system in offspring. These studies reported several

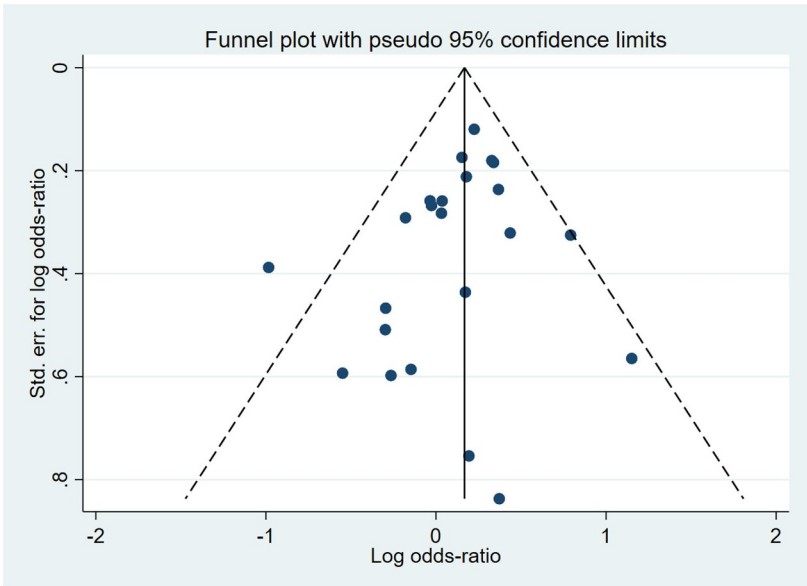

**Fig 8. Funnel plot of the association between maternal SSRIs exposure and risk of congenital kidney and urinary tract abnormalities in offspring.**

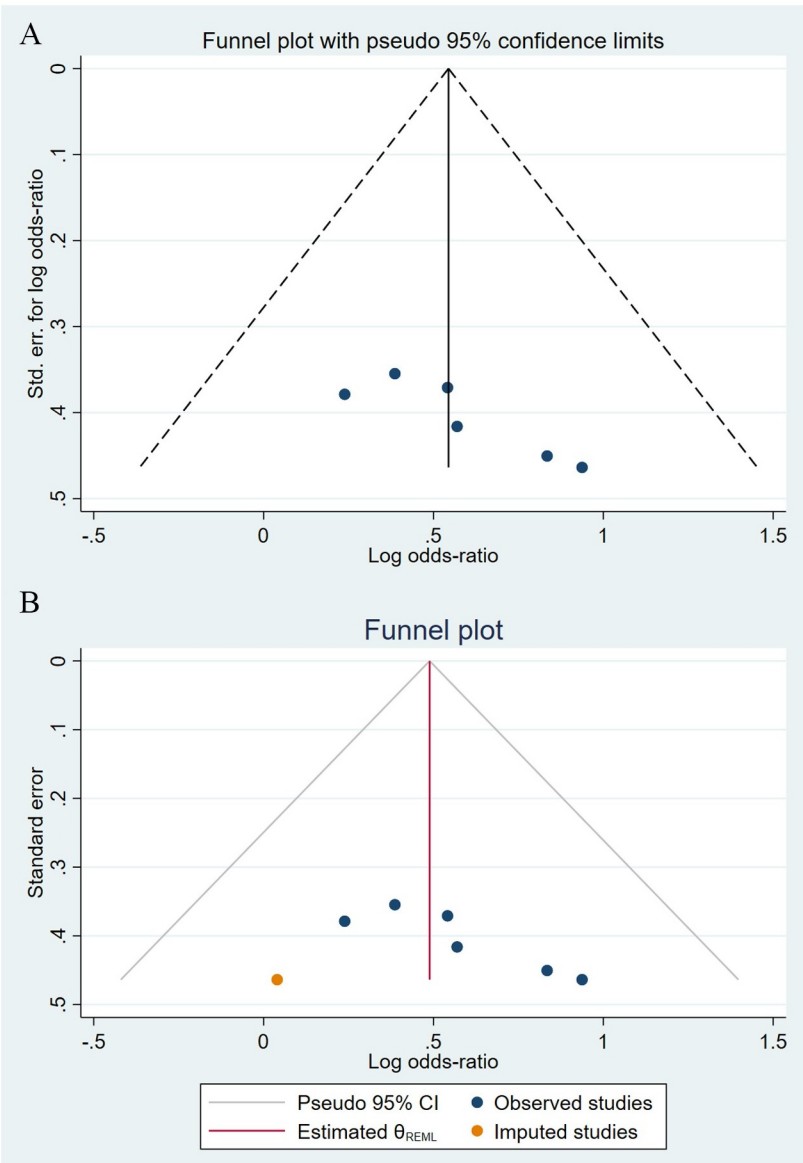

**Fig 9. Funnel plot of the association between maternal SNRIs exposure and risk of congenital kidney and urinary tract abnormalities in offspring.** A: Funnel plot before trim-and-fill methods. B: Funnel plot after trim-and-fill methods.

common congenital anomalies of the nervous system, including neural tube defects, spina bifida, anencephaly, hydrocephalus, Dandy–Walker malformation, cerebellar hypoplasia and holoprosencephaly.

Compared to the offspring of women not exposed to SNRIs, the offspring of women exposed to SNRIs were at higher risk of nervous system malformations. However, no association was observed in SSRIs group. The pooled OR was 1.07 (95% CI: 0.93, 1.23) for those women exposed to SSRIs during pregnancy, with the reported ORs ranging from 0.57 (95% CI: 0.23, 1.39) to 3.63 (95% CI: 0.51, 25.78) Fig 10. The pooled OR was 2.28 (95% CI: 1.50, 3.45) for those women exposed to SNRIs during pregnancy, with the reported ORs ranging

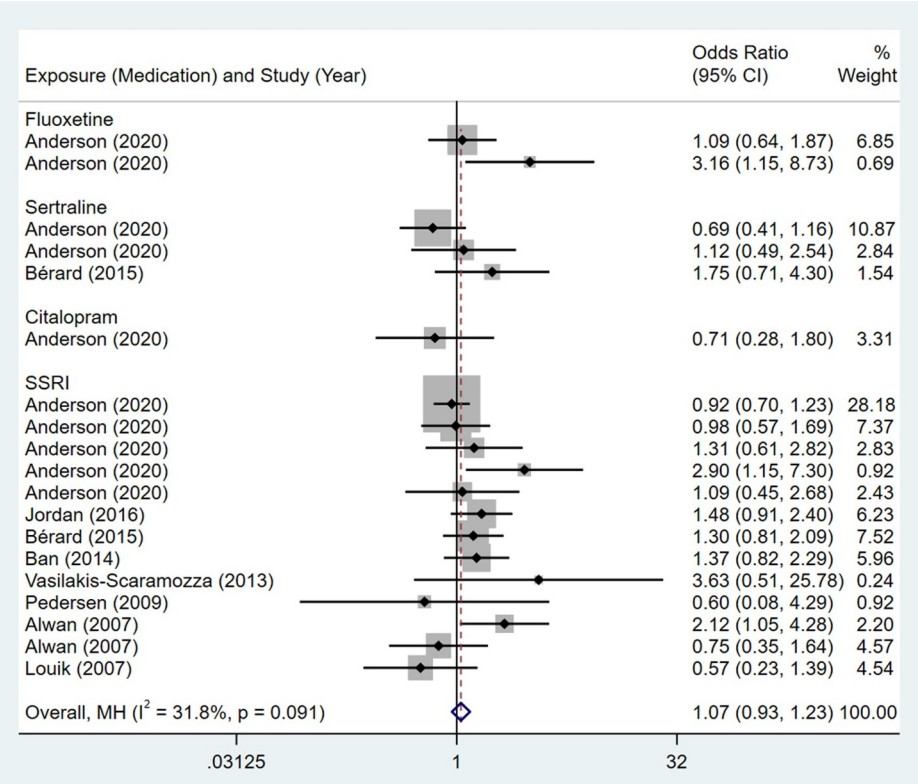

**Fig 10. Forest plot of the association between maternal SSRIs exposure and risk of congenital nervous system anomalies in offspring.**

from 1.64 (95% CI: 0.53, 5.08) to 5.61 (95% CI: 1.84, 17.11) Fig 11. No significant heterogeneity was detected in either group ($I^2$ = 31.8%, P = 0.091; $I^2$ = 0.0%, P = 0.562). For those women exposed to SSRIs, no publication bias was detected as indicated by the funnel plot and Egger's test (Coefficient = 0.85, 95% CI: -0.53, 2.22; P = 0.212). For those women exposed to SNRIs, no publication bias was detected either based on the funnel plot and Egger's test (Coefficient = 0.73, 95% CI: -5.62, 7.07; P = 0.740) Fig 12.

## Risk of congenital anomalies of the digestive system

Nine studies assessed maternal exposure to SSRIs and the risk of congenital digestive malformations in offspring, and three studies assessed maternal exposure to SNRIs and the risk of congenital digestive malformations in offspring. These studies reported the congenital anomalies of the tubular gastrointestinal tract such as esophageal atresia, intestinal atresia, anorectal atresia and Hirschsprung's Disease, as well as cleft lip with or without cleft palate.

Compared to the offspring of women not exposed to SSRIs or SNRIs, the offspring of women exposed to SSRIs or SNRIs were at higher risk of digestive system anomalies. The pooled OR was 1.11 (95% CI: 1.01, 1.21) for those women exposed to SSRIs during pregnancy, with the reported ORs ranging from 0.29 (95% CI: 0.04, 2.15) to 2.81 (95% CI: 1.33, 5.92) Fig 13. The pooled OR was 2.05 (95% CI: 1.60, 2.64) for those women exposed to SNRIs during pregnancy, with the reported ORs ranging from 0.78 (95% CI: 0.19, 3.11) to 3.58 (95% CI: 1.44, 8.89) Fig 14. No significant heterogeneity was detected in either group ($I^2$ = 0.0%,

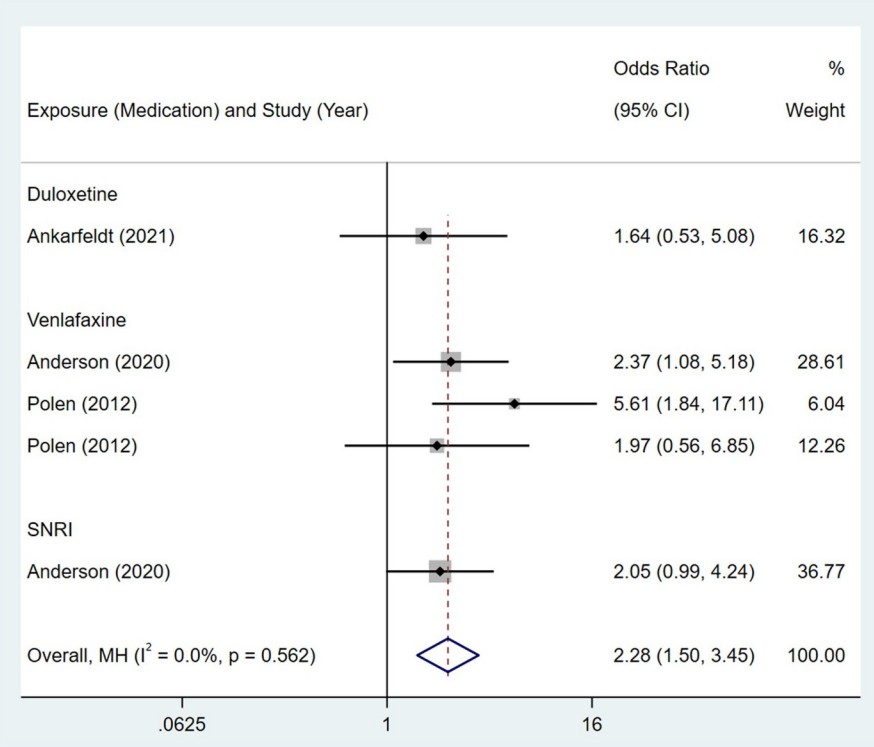

**Fig 11. Forest plot of the association between maternal SNRIs exposure and risk of congenital nervous system anomalies in offspring.**

P = 0.614; $I^2$ = 0.0%, P = 0.428). For those women exposed to SSRIs, no publication bias was detected as indicated by the funnel plot and Egger's test (Coefficient = 0.42, 95% CI: -0.19, 1.03; P = 0.176). For those women exposed to SNRIs, no publication bias was detected either based on the funnel plot and Egger's test (Coefficient = -0.29, 95% CI: -3.60, 3.02; P = 0.818) Fig 15.

## Risk of abdominal birth defects

Seven studies assessed maternal exposure to SSRIs and the risk of congenital abdominal birth defects in offspring, and four studies assessed maternal exposure to SNRIs and the risk of congenital abdominal birth defects in offspring. These studies reported three types of birth defects that affect the abdomen: diaphragmatic hernia (organs protrude into the chest cavity), omphalocele (organs protrude through the navel) and gastroschisis (organs protrude through the abdominal wall).

Compared to the offspring of women not exposed to SSRIs or SNRIs, the offspring of women exposed to SSRIs or SNRIs were at higher risk of abdominal birth defects. The pooled OR was 1.33 (95% CI: 1.16, 1.53) for those women exposed to SSRIs during pregnancy, with the reported ORs ranging from 0.86 (95% CI: 0.27, 2.74) to 3.32 (95% CI: 1.55, 7.08) Fig 16. The pooled OR was 2.91 (95% CI: 1.98, 4.28) for those women exposed to SNRIs during pregnancy, with the reported ORs ranging from 1.93 (95% CI: 0.44, 8.50) to 5.17 (95% CI: 1.29, 20.76) Fig 17. No significant heterogeneity was detected in either group ($I^2$ = 18.1%, P = 0.233; $I^2$ = 0.0%, P = 0.920). For those women exposed to SSRIs, no publication bias was detected as

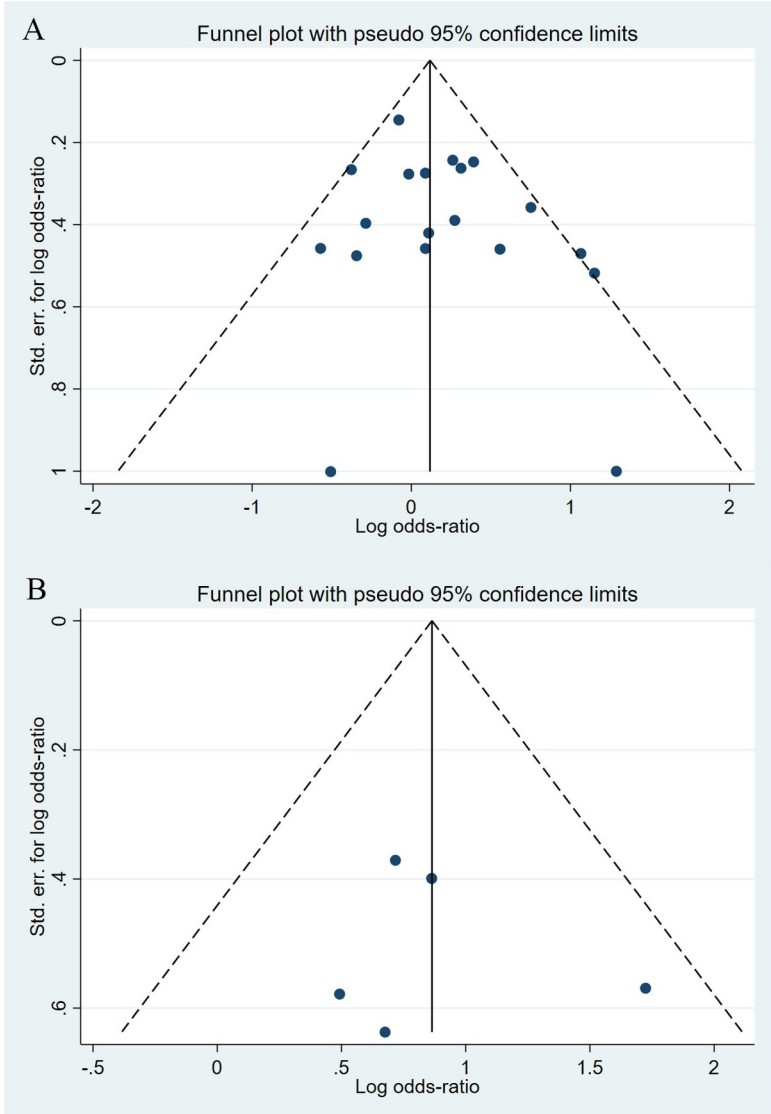

**Fig 12. Funnel plot of the association between maternal SSRIs/SNRIs exposure and risk of congenital nervous system anomalies in offspring.** A: Funnel plot for SSRIs. B: Funnel plot for SNRIs.

indicated by the funnel plot and Egger's test (Coefficient = 0.53, 95% CI: -0.88, 1.95; P = 0.437). For those women exposed to SNRIs, no publication bias was detected either based on the funnel plot and Egger's test (Coefficient = 0.414, 95% CI: -2.40, 3.23; P = 0.704) Fig 18.

## Risk of musculoskeletal congenital malformations

Eleven studies assessed maternal exposure to SSRIs and the risk of musculoskeletal congenital malformations in offspring, and three studies assessed maternal exposure to SNRIs and the risk of musculoskeletal congenital malformations in offspring. These studies reported musculoskeletal congenital malformations: craniosynostosis, limb-reduction defects, clubfoot and talipes equinovarus.

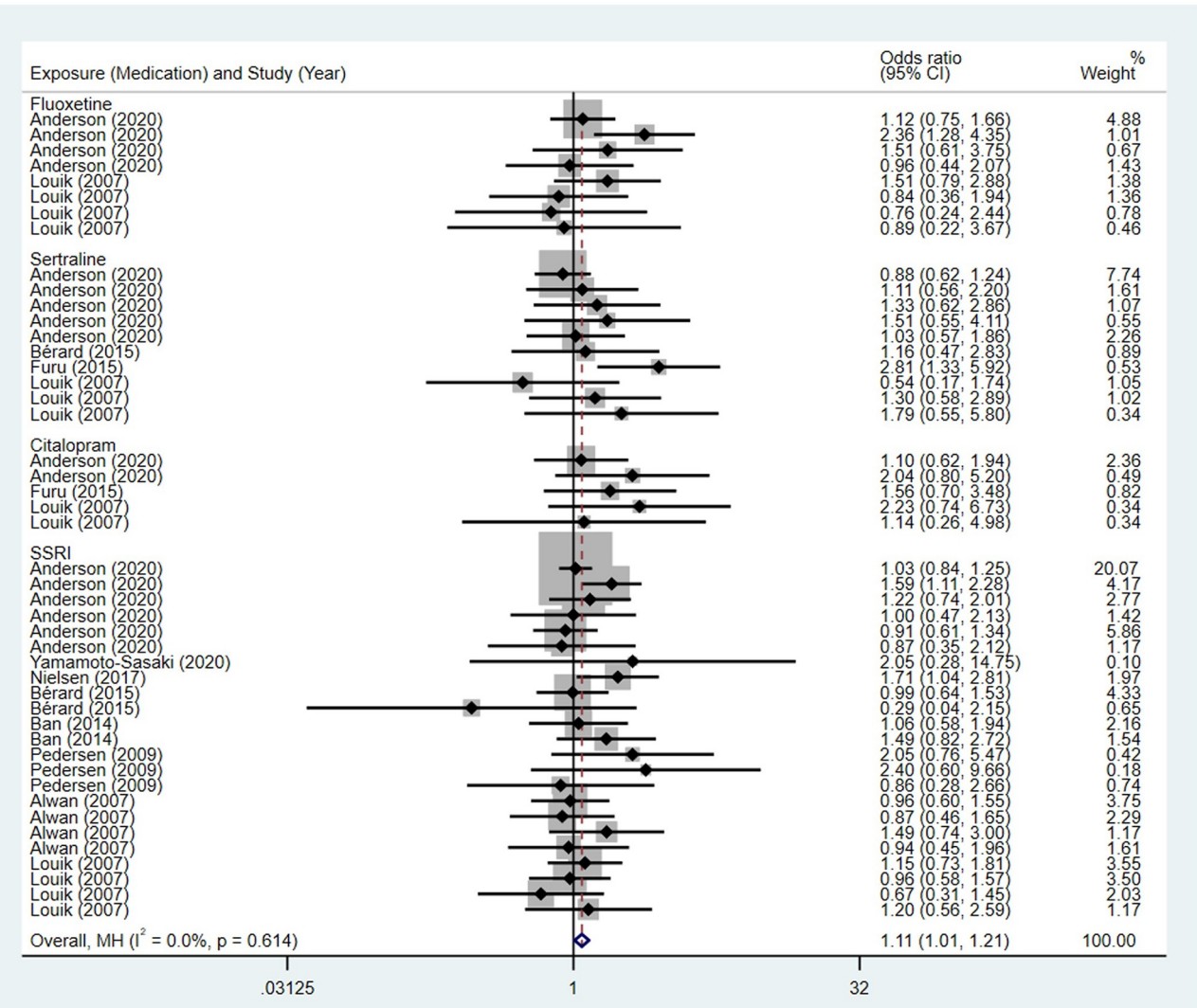

**Fig 13. Forest plot of the association between maternal SSRIs exposure and risk of congenital digestive system anomalies in offspring.**

Compared to the offspring of women not exposed to SSRIs, the offspring of women exposed to SSRIs were at higher risk of musculoskeletal congenital malformations. However, no association was observed in SNRIs group. The pooled OR was 1.44 (95% CI: 1.32, 1.56) for those women exposed to SSRIs during pregnancy, with the reported ORs ranging from 0.37 (95% CI: 0.09, 1.50) to 4.44 (95% CI: 1.10, 18.01), and modest heterogeneity was detected ($I^2$ = 43.1%, P = 0.003) Fig 19. The pooled OR was 1.01 (95% CI: 0.73, 1.40) for those women exposed to SNRIs during pregnancy, with the reported ORs ranging from 0.81 (95% CI: 0.54, 1.23) to 2.03 (95% CI: 0.58, 7.07), and no significant heterogeneity was detected ($I^2$ = 14.5%, P = 0.322) Fig 20. For those women exposed to SSRIs, no publication bias was detected as indicated by the funnel plot and Egger's test (Coefficient = -0.28, 95% CI: -1.16, 0.60; P = 0.522) Fig 21. For those women exposed to SNRIs, publication bias may exist as indicated by the funnel plot and Egger's test (Coefficient = 2.05, 95% CI: 1.00, 3.09; P = 0.008). When we used the trim and fill method to adjust for the potential publication bias, the recalculated pooled odds

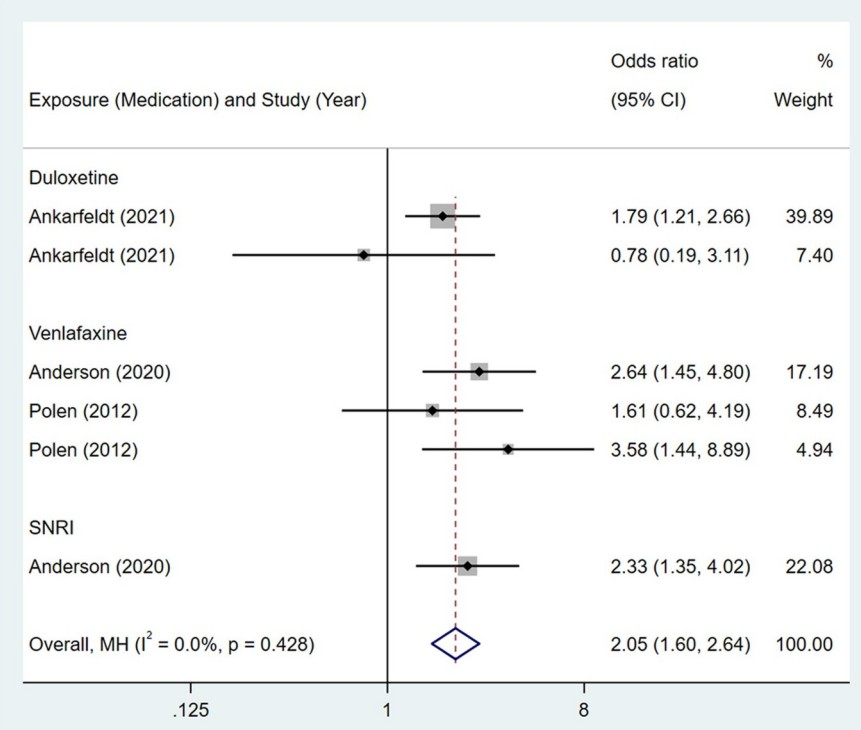

**Fig 14. Forest plot of the association between maternal SNRIs exposure and risk of congenital digestive system anomalies in offspring.**

ratio of association of maternal SNRIs exposure with musculoskeletal congenital malformations in offspring was 0.90 (95% CI: 0.60, 1.36) Fig 22.

## Risk of congenital malformations of eye, ear, face and neck

Six studies assessed maternal exposure to SSRIs and the risk of congenital malformations of eye, ear, face and neck in offspring, and two studies assessed maternal exposure to SNRIs and the risk of congenital malformations of eye, ear, face and neck in offspring. These studies reported several congenital malformations such as anotia, microtia, cataracts, glaucoma and anterior chamber defects.

No association was observed between maternal exposure to SSRIs or SNRIs and congenital malformations of eye, ear, face and neck in offspring. The pooled OR was 1.00 (95% CI: 0.80, 1.26) for those women exposed to SSRIs during pregnancy, with the reported ORs ranging from 0.47 (95% CI: 0.07, 3.39) to 3.13 (95% CI: 1.36, 7.19) Fig 23. The pooled OR was 1.44 (95% CI: 0.82, 2.55) for those women exposed to SNRIs during pregnancy, with the reported ORs ranging from 1.22 (95% CI: 0.16, 9.27) to 1.68 (95% CI: 0.75, 3.74) Fig 24. No significant heterogeneity was detected in either group ($I^2$ = 26.0%, P = 0.182; $I^2$ = 0.0%, P = 0.885). For those women exposed to SSRIs, no publication bias was detected as indicated by the funnel plot and Egger's test (Coefficient = -0.67, 95% CI: -2.54, 1.20; P = 0.447). For those women exposed to SNRIs, no publication bias was detected either based on the funnel plot and Egger's test (Coefficient = -0.41, 95% CI: -10.16, 9.35; P = 0.689) Fig 25.

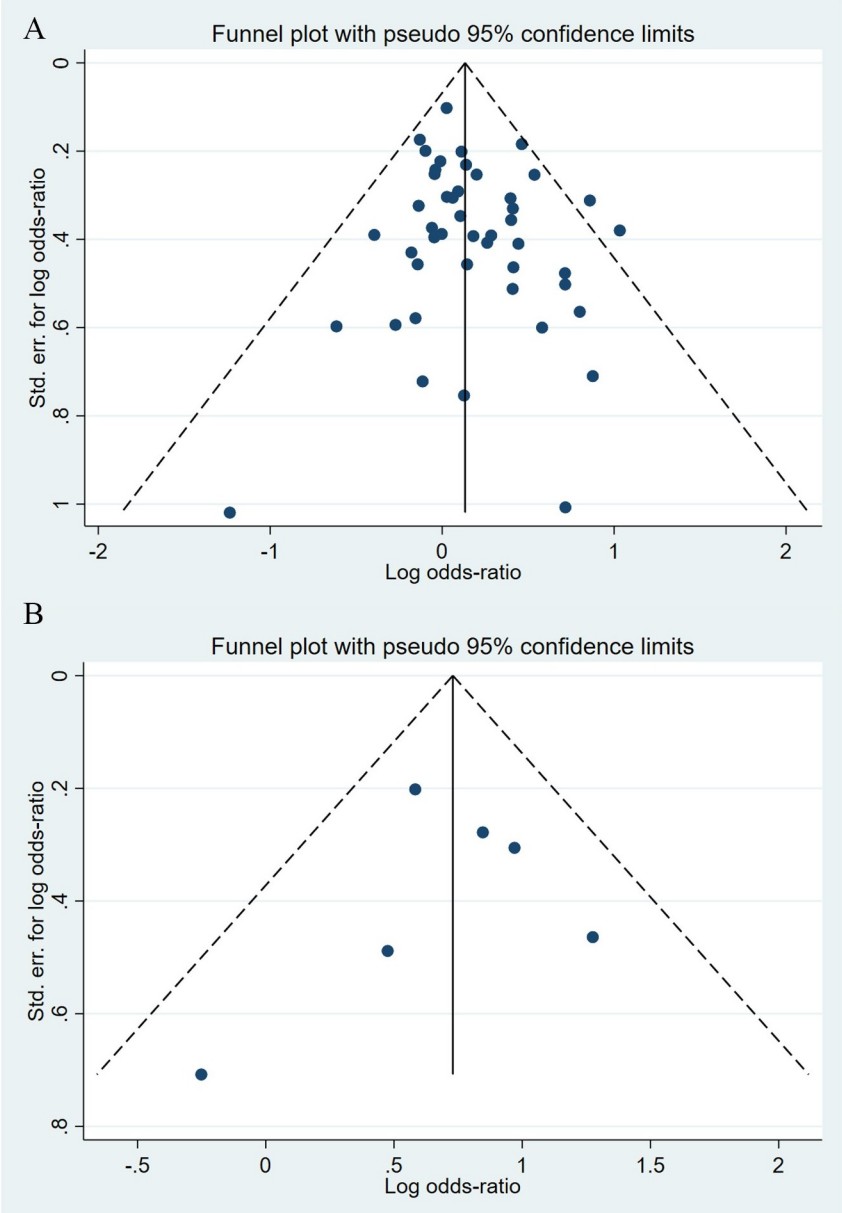

**Fig 15. Funnel plot of the association between maternal SSRIs/SNRIs exposure and risk of congenital digestive system anomalies in offspring.** A: Funnel plot for SSRIs. B: Funnel plot for SNRIs.

## Risk of congenital malformations of genital organs

Four studies assessed maternal exposure to SSRIs exposure and the risk of congenital malformations of genital organs in offspring, and two studies assessed maternal exposure to SNRIs and the risk of congenital malformations of genital organs in offspring. One study reported undescended testis, and the remaining studies did not specify the categories of congenital malformations of genital organs.

No association was observed between maternal exposure to SSRIs or SNRIs and congenital malformations of genital organs in offspring. The pooled OR was 0.84 (95% CI: 0.65, 1.07) for

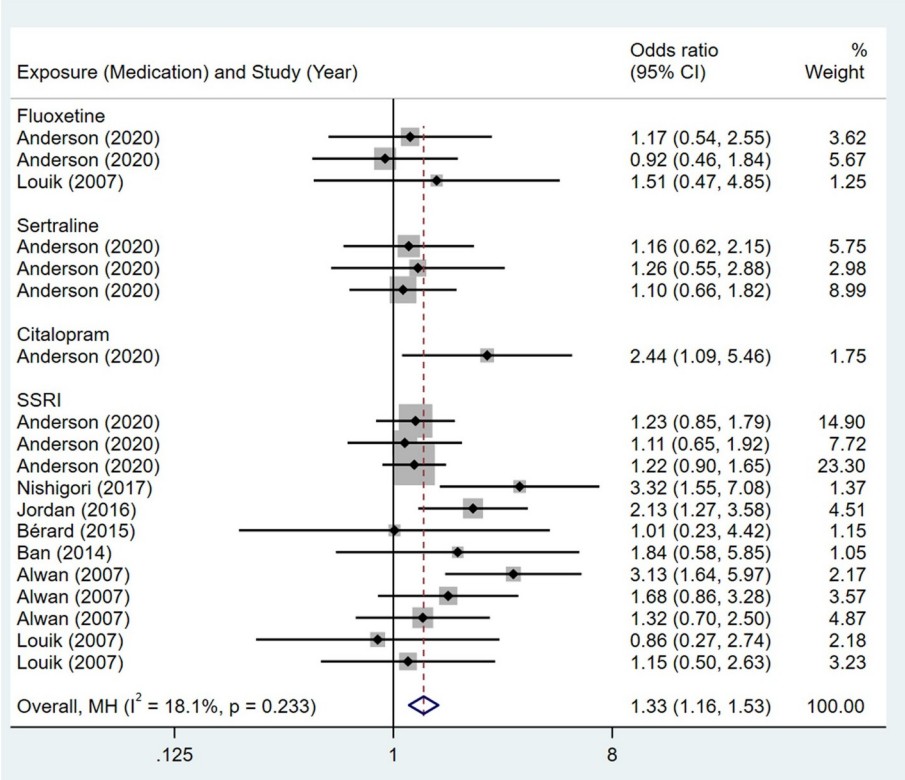

**Fig 16. Forest plot of the association between maternal SSRIs exposure and risk of congenital abdominal birth defects in offspring.**

those women exposed to SSRIs during pregnancy, with the reported ORs ranging from 0.72 (95% CI: 0.39, 1.35) to 2.25 (95% CI: 0.51, 9.86) Fig 26. The pooled OR was 1.01 (95% CI: 0.48, 2.13) for those women exposed to SNRIs during pregnancy, with the reported ORs from two studies: 0.91 (95% CI: 0.41, 2.04) and 2.98 (95% CI: 0.41, 21.78) Fig 27. No significant heterogeneity was detected in either group ($I^2 = 0.0\%$, P = 0.606; $I^2 = 15.9\%$, P = 0.275). For those women exposed to SSRIs, no publication bias was detected as indicated by the funnel plot and Egger's test (Coefficient = 1.72, 95% CI: -0.23, 3.68; P = 0.071) Fig 28. For those women exposed to SNRIs, publication bias may exist as indicated by the funnel plot. When we used the trim and fill method to adjust for the potential publication bias, the recalculated pooled odds ratio of the association of maternal SNRIs exposure with malformations of genital organs in offspring was 0.91 (95% CI: 0.46, 1.84) Fig 29.

## Risk of congenital malformations of the respiratory system

Three studies assessed maternal exposure to SSRIs and the risk of congenital malformations of the respiratory system in offspring, and one study assessed the maternal exposure to SNRIs and the risk of congenital malformations of the respiratory system in offspring. One study reported choanal atresia, and the remaining studies did not specify the categories of congenital malformations of the respiratory system.

No association was observed between maternal exposure to SSRIs or SNRIs and congenital malformations of the respiratory system in offspring. The pooled OR was 1.22 (95%CI: 0.80,

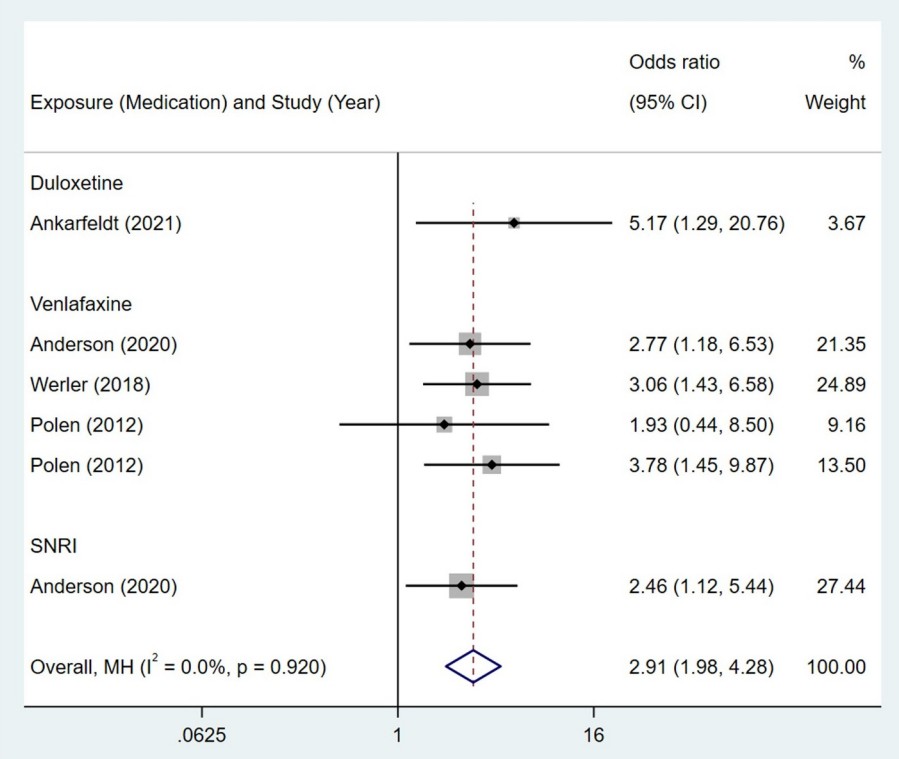

**Fig 17. Forest plot of the association between maternal SNRIs exposure and risk of congenital abdominal birth defects in offspring.**

1.84) for those women exposed to SSRIs during pregnancy, with the reported ORs ranging from 0.75 (95%CI: 0.36, 1.55) to 1.75 (95%CI: 0.85, 3.59). No significant heterogeneity was detected ($I^2$ = 41.7%, P = 0.180), and no publication bias was detected as indicated by the funnel plot and Egger's test (Coefficient = -64.74, 95% CI: -789.10, 659.61; P = 0.460) Fig 30. Only one study reported the maternal SNRIs exposure and the risk of congenital malformations of the respiratory system in offspring with OR of 1.51 (95% CI: 0.49, 4.70).

## Overall malformation risk

We analyzed 14 studies on SSRIs that reported the case number of major malformation, and 8 studies on SNRIs that reported the case number of major malformation. Major malformation refers to the occurrence of at least one of the nine malformations we have listed previously. Of the total of 22 studies, 17 studies are from Table 1, and the remaining five studies did not classify malformations and were not included in our previous analyses [52–56].

Compared to the offspring of women not exposed to SSRIs, the offspring of women exposed to SSRIs had a higher overall malformation rate. Similarly, the offspring of women exposed to SNRIs had a higher overall malformation rate when compared to the offspring of women not exposed to SNRIs. The pooled OR between major malformation and exposure to SSRIs was 1.17 (95% CI: 1.13, 1.21), with the reported ORs ranging from 0.90 (95% CI: 0.66, 1.23) to 1.49 (95% CI: 1.22, 1.81), and no significant heterogeneity was detected ($I^2$ = 35.4%, P = 0.093) Fig 31. The pooled OR between major malformation and exposure to SNRIs was 1.27 (95% CI:

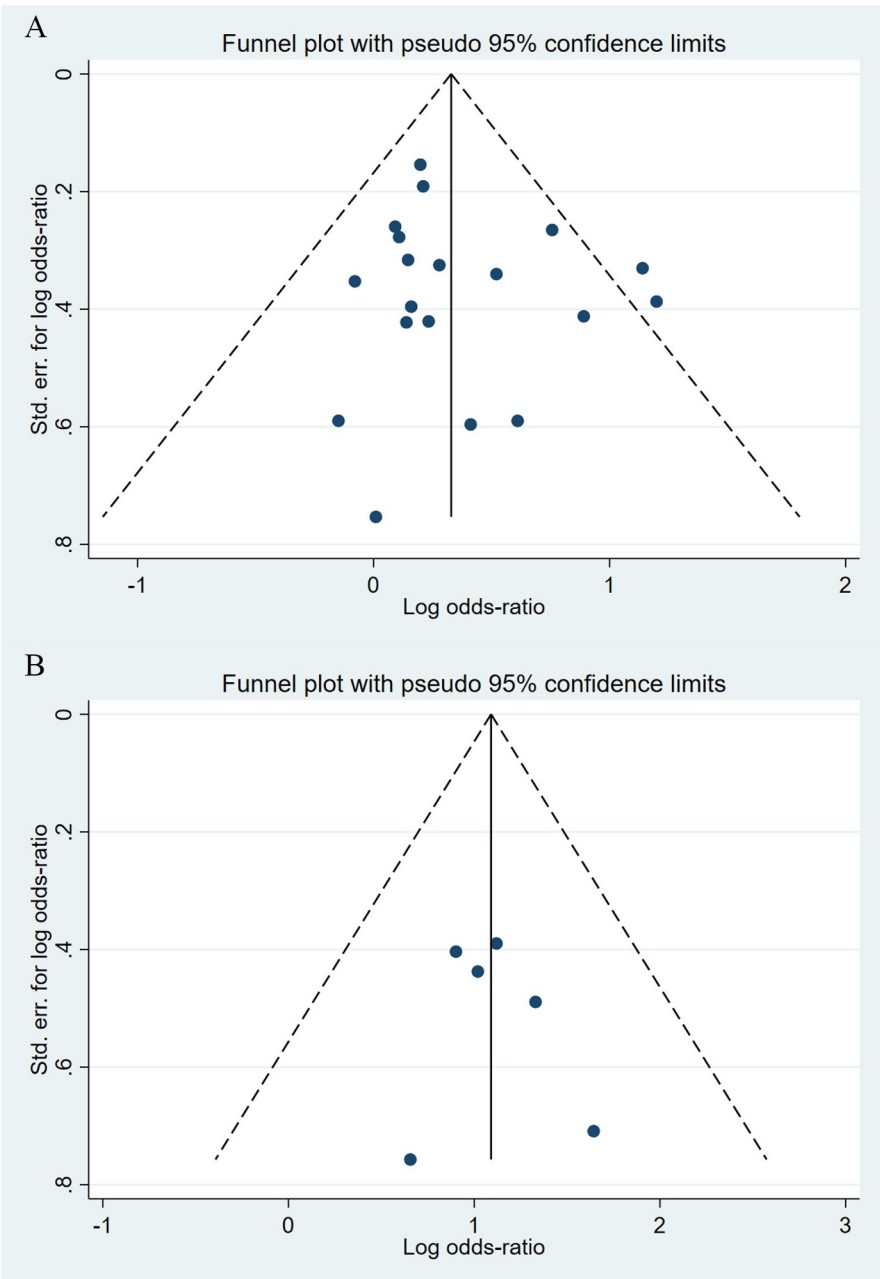

**Fig 18. Funnel plot of the association between maternal SSRIs/SNRIs exposure and risk of congenital abdominal birth defects in offspring.** A: Funnel plot for SSRIs. B: Funnel plot for SNRIs.

1.15, 1.40), with the reported ORs ranging from 1.05 (95% CI: 0.35, 3.11) to 2.32 (95% CI: 1.31, 4.10), and no significant heterogeneity was detected ($I^2$ = 49.2%, P = 0.055) Fig 32. No publication bias was detected for women exposed to SSRIs, as indicated by the funnel plot and Egger's test (Coefficient = -0.05, 95% CI: -1.39, 1.29; P = 0.934). For those women exposed to SNRIs, no publication bias was detected either based on the funnel plot and Egger's test (Coefficient = 0.95, 95% CI: -1.43, 3.33; P = 0.366) Fig 33.

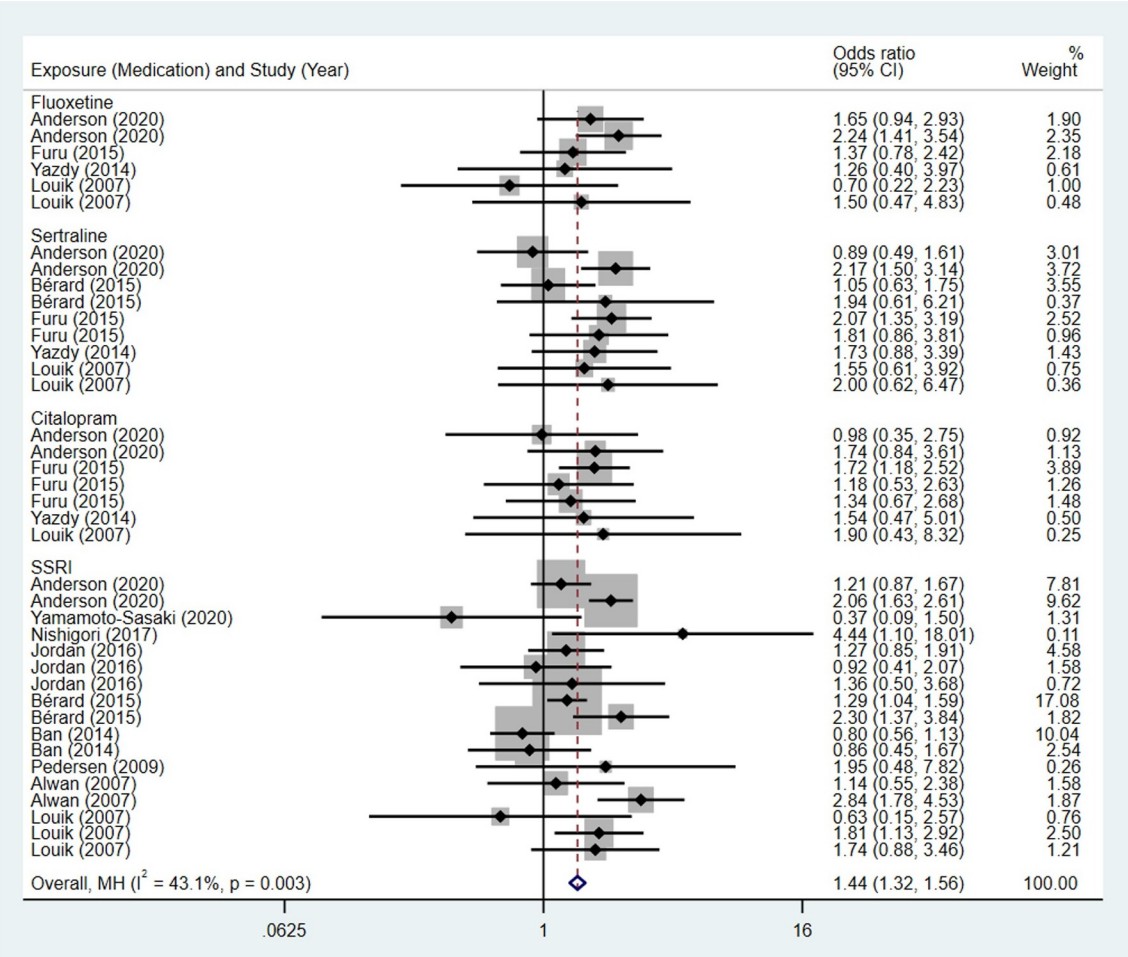

**Fig 19. Forest plot of the association between maternal SSRIs exposure and risk of musculoskeletal congenital malformations in offspring.**

## Discussion

To our knowledge, this is the first meta-analysis to examine the association of maternal exposure to SNRIs during pregnancy with system-specific malformations in offspring and the first to examine the difference in this association compared to SSRIs. We found that maternal exposure to SNRIs may be associated with a higher risk of congenital cardiovascular abnormalities, anomalies of the kidney and urinary tract, malformations of nervous system, anomalies of the digestive system and abdominal birth defects in offspring, but such association was not observed for other types of congenital abnormalities such as congenital musculoskeletal malformations, malformations of genital organs, malformations of the respiratory system, and malformations of eye, ear, face and neck. On the other hand, we found that maternal exposure to SSRIs during pregnancy may be associated with higher risk of congenital cardiovascular abnormalities, anomalies of the kidney and urinary tract, anomalies of the digestive system, abdominal birth defects and musculoskeletal malformations in offspring, but such association was not observed for congenital malformations of the nervous system, malformations of genital organs, malformations of the respiratory system, and malformations of eye, ear, face and

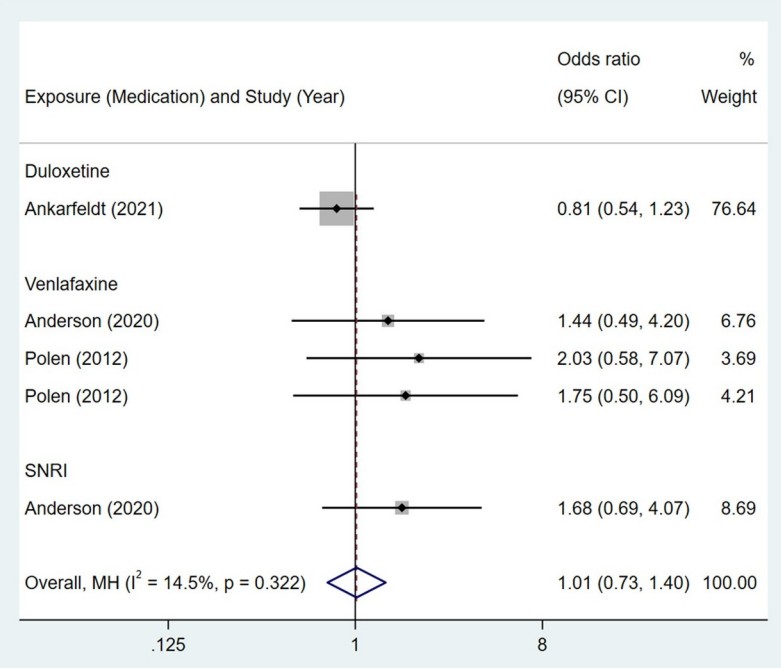

**Fig 20. Forest plot of the association between maternal SNRIs exposure and risk of musculoskeletal congenital malformations in offspring.**

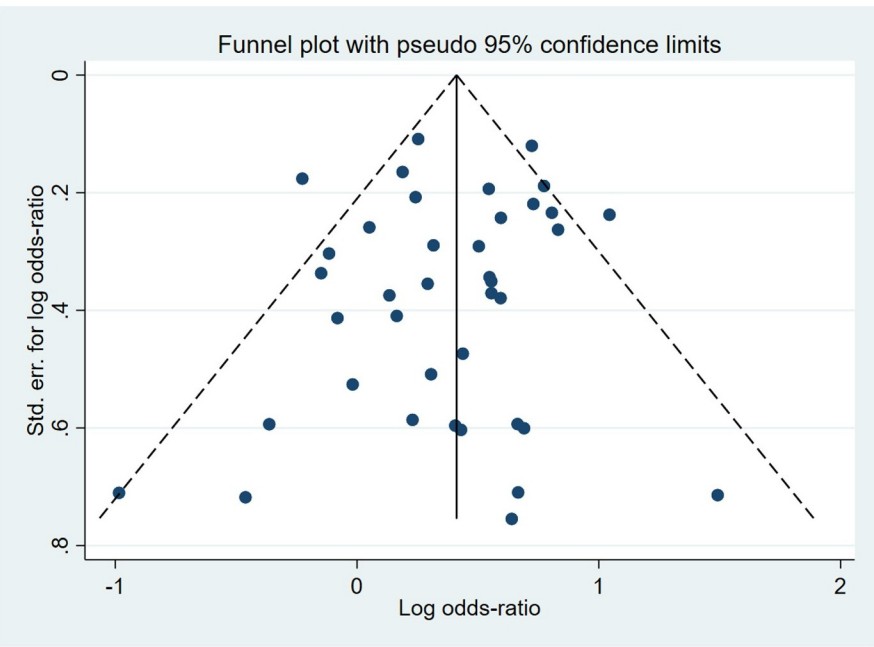

**Fig 21. Funnel plot of the association between maternal SSRIs exposure and risk of musculoskeletal congenital malformations in offspring.**

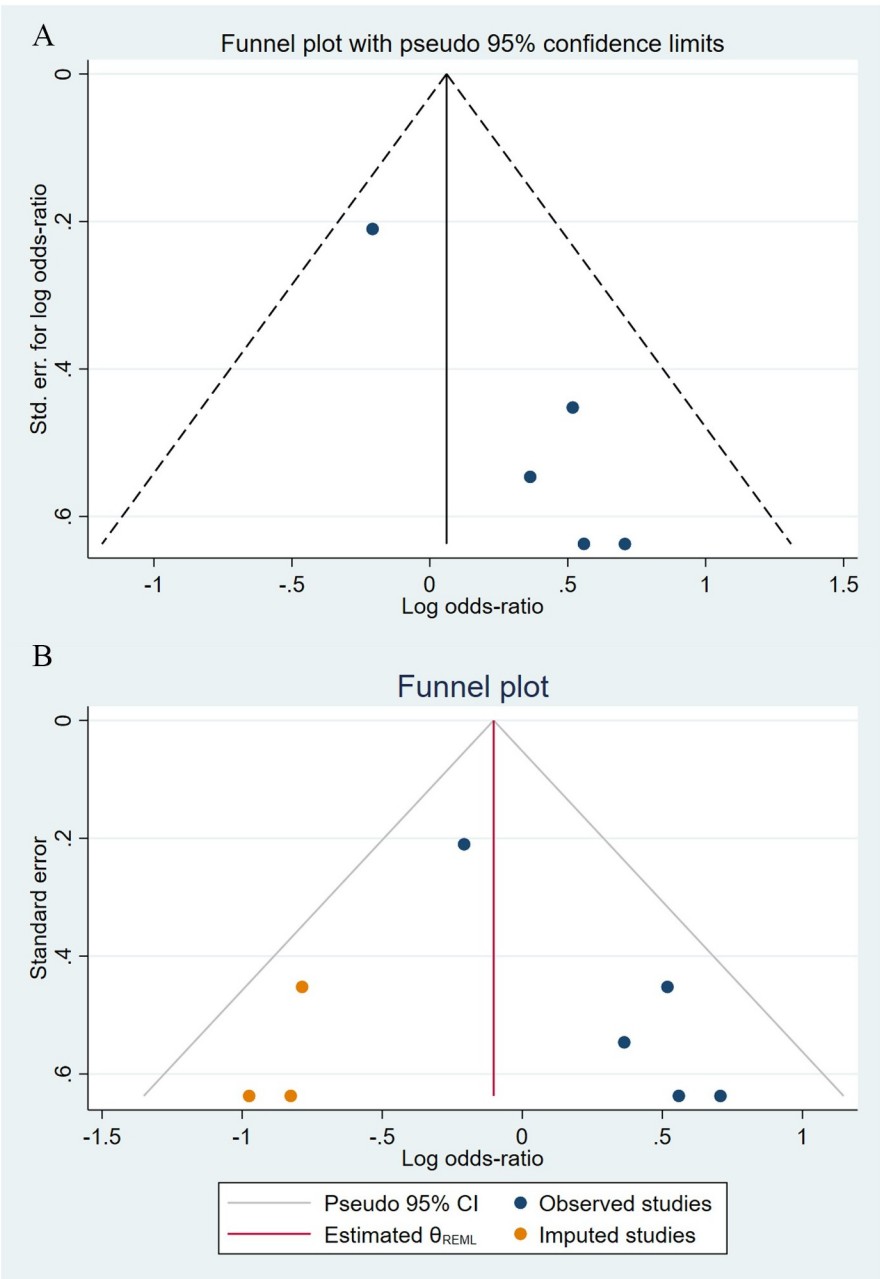

**Fig 22. Funnel plot of the association between maternal SNRIs exposure and risk of musculoskeletal congenital malformations in offspring.** A: Funnel plot before trim-and-fill methods. B: Funnel plot after trim-and-fill methods.

neck. Regarding the overall malformation risk, offspring exposed to SSRIs or SNRIs also had a higher risk of congenital malformations than non-exposed offspring. Moreover, maternal exposure to SNRIs may have a larger teratogenic risk in offspring compared to maternal exposure to SSRIs for the congenital malformations associated with both exposures, including congenital cardiovascular abnormalities, anomalies of the digestive system and abdominal birth defects. However, when it came to the overall malformation risk, this risk difference was less prominent.

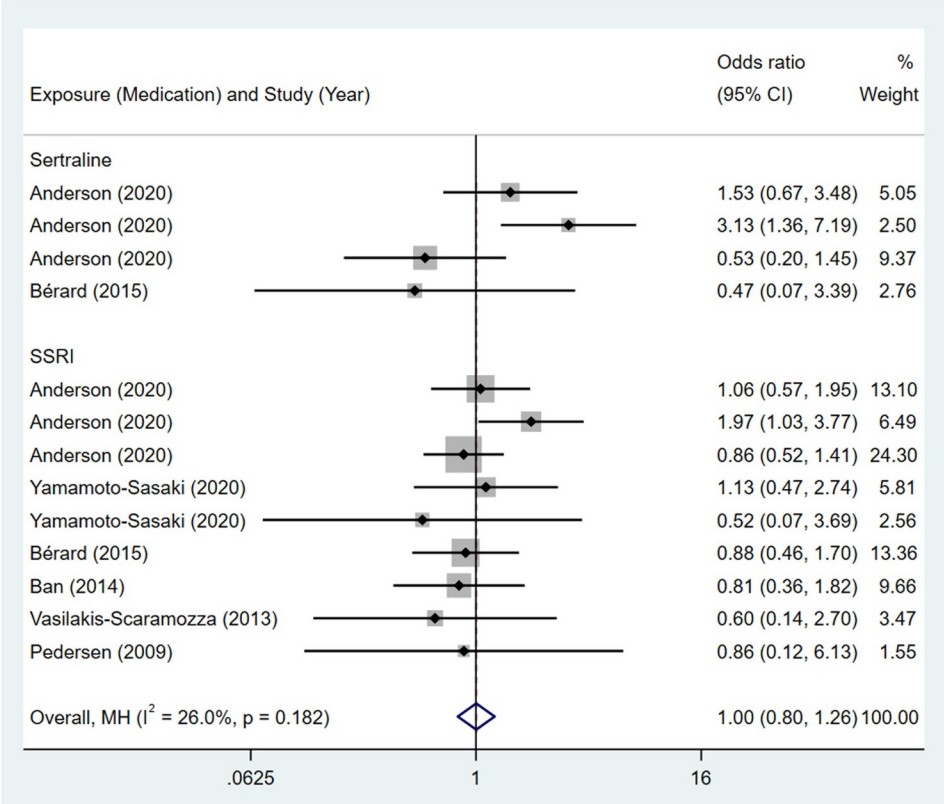

**Fig 23. Forest plot of the association between maternal SSRIs exposure and risk of congenital malformations of eye, ear, face and neck in offspring.**

This meta-analysis supports the hypothesis that maternal exposure to antidepressants (SSRIs or SSNIs) during pregnancy has a negative influence on fetal development, even if research on the association of exposure to these medications with congenital abnormalities in offspring has produced inconsistent results [25, 26]. Our findings are consistent with previous meta-analyses studies: maternal exposure to SSRIs is associated with a higher risk of congenital cardiovascular abnormalities in offspring, although this teratogenic effect is not substantial [18, 57, 58]. Many previous studies have demonstrated adverse cardiac effects in animal models with disrupted serotonin signaling, from either serotonin reuptake inhibitor or serotonin-norepinephrine reuptake inhibitor exposure during development [59–62]. Furthermore, this association has been demonstrated at the level of gene expression in human placental tissues [63]. Of note, we did not include studies on Paroxetine in the meta-analysis because in 2005 the FDA issued a Public Health Advisory and reclassified Paroxetine as pregnancy-category D which means that there is evidence of risk to the fetus with the use of this medication, and it is not recommended for use during pregnancy by the American College of Obstetricians and Gynecologists (ACOG). If the studies on Paroxetine were included in the meta-analysis, the pooled ORs would most likely increase because many studies have reported a substantial association of maternal exposure to Paroxetine during pregnancy with the higher risk of congenital abnormalities in offspring [64, 65].

In addition to cardiovascular malformations, we also found that maternal exposure to SSRI or SSNI was associated with a higher risk of congenital anomalies of the kidney and urinary

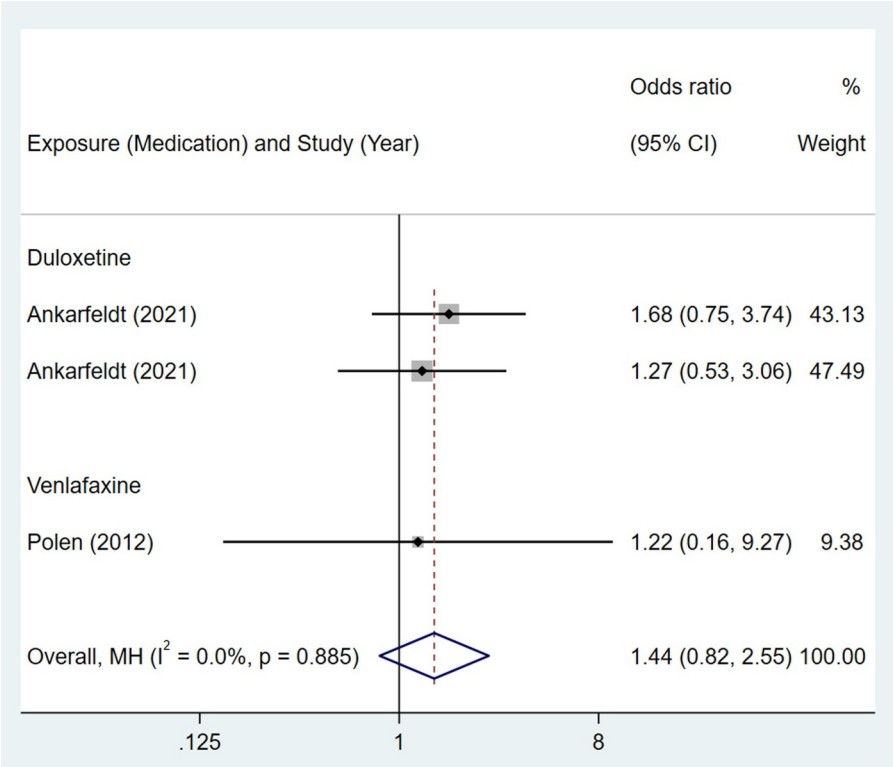

**Fig 24. Forest plot of the association between maternal SNRIs exposure and risk of congenital malformations of eye, ear, face and neck in offspring.**

tract, malformations of the nervous system, anomalies of the digestive system and musculo-skeletal malformations in offspring. Although there are few meta-analyses of these associations [5, 18], the biological mechanisms supporting these associations have been demonstrated in animal models [66]. Some research has demonstrated the potential mechanisms through which maternal exposure to SSRIs had a negative effect on skeletal abnormalities in offspring such as affecting cell cycle [67] and affecting osteoblast maturation [68]. Previous study has demonstrated that prenatal exposure to Venlafaxine induced oxidative stress which caused apoptotic neurodegeneration in striatum and hippocampus of developing fetal brain [69]. Maternal exposure to Desvenlafaxine was demonstrated to increase the expression of nerve growth factor (NGF), brain-derived neurotrophic factor (BDNF) and S100b in the fetal brain, altering brain development of albino rats [70]. Prowse et al. [71] found that maternal exposure to SSRIs had structural consequences on the developing enteric nervous system (ENS) in the exposed offspring of Wistar rats, influencing the development of the gastrointestinal tract of the offspring. The influence of maternal exposure to SSRIs on kidney development in rat neonates was demonstrated in a study: Ghavamabadi et al. [72] found that maternal exposure to Fluoxetine is associated with the neonatal kidney developmental deficiency by reducing the gene expression of BMP7 and WNT4 in the kidney.

There have been studies of the difference in the risk of adverse clinical outcomes between SSRIs and SNRIs [16, 17], and we found that this risk difference may also exist in congenital abnormalities in exposed offspring. In this meta-analysis, we found that the offspring of women exposed to SNRIs had a higher risk of congenital cardiovascular abnormalities, anomalies of the digestive system and abdominal birth defects, compared to the offspring of women

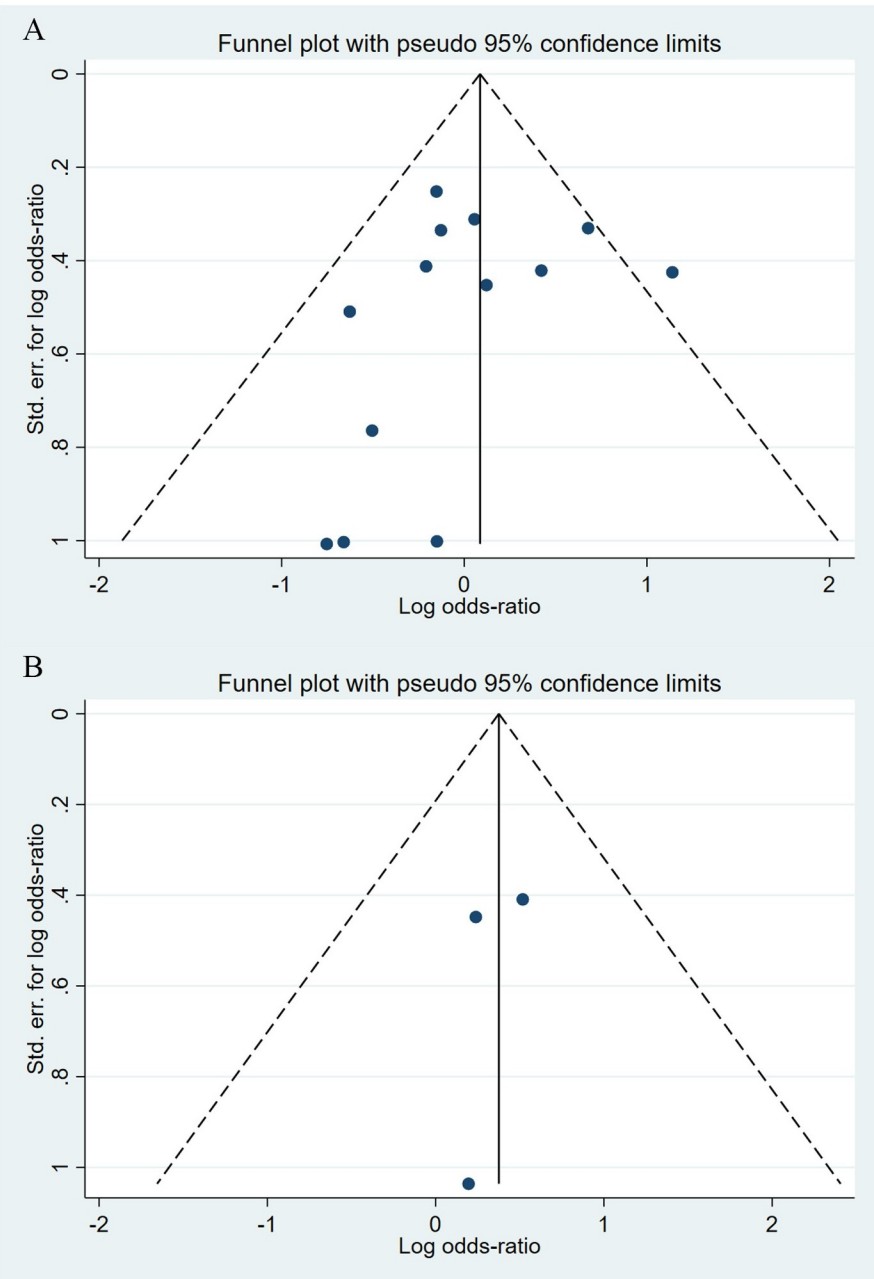

**Fig 25. Funnel plot of the association between maternal SSRIs/SNRIs exposure and risk of congenital malformations of eye, ear, face and neck in offspring.** A: Funnel plot for SSRIs. B: Funnel plot for SNRIs.

exposed to SSRIs Table 2. This risk difference may also exist in congenital anomalies of the kidney and urinary tract, although the results showed that the difference was not statistically significant. In addition, according to data on congenital malformations published by the Centers for Disease Control and Prevention (CDC), the cardiovascular abnormalities accounted for half of the ten congenital abnormalities with the highest prevalence, followed by the anomalies of the digestive system Table 2. Our study found that the risk of congenital cardiovascular abnormalities and anomalies of the digestive system differed between SSRIs and SNRIs, so it is

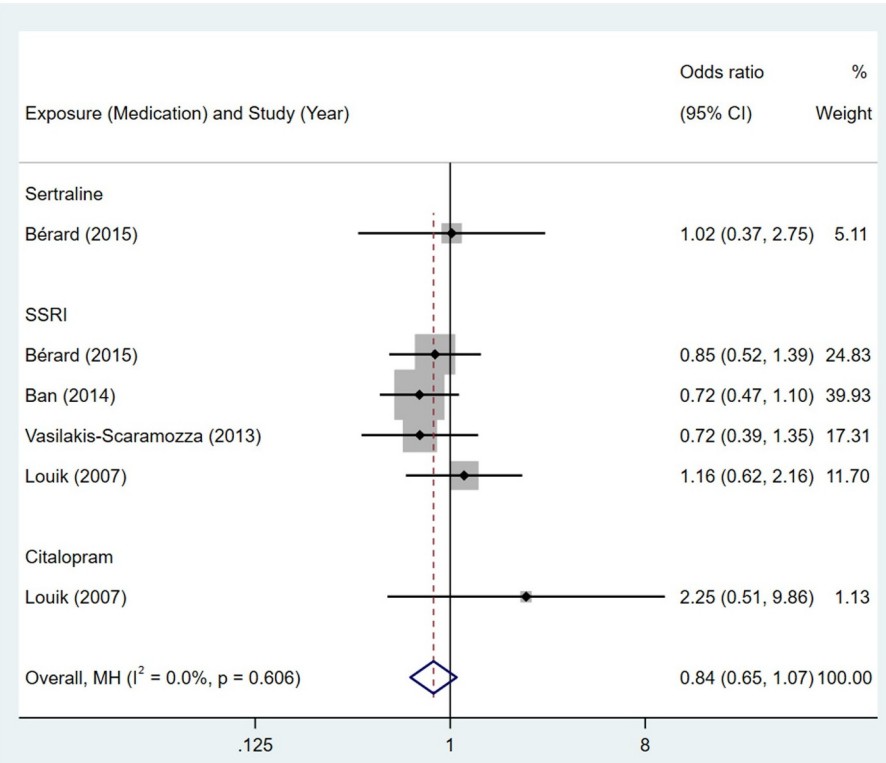

**Fig 26. Forest plot of the association between maternal SSRIs exposure and risk of congenital malformations of genital organs in offspring.**

critical to distinguish their differences in teratogenic effects to guide clinical management. SSRIs block the reuptake of serotonin, while SNRIs block the reuptake of both serotonin and norepinephrine. Perhaps this difference in the blocking of neurotransmitters' reuptake leads to the differences in their pharmacological effects, which causes the difference in the risk of adverse clinical outcomes. We did not find the potential biological mechanisms to explain these results based on current literature but hope our findings can spark new ideas for future scientific research. In addition, we did not examine this risk difference among individual medications, but we cannot rule out the possibility of this difference among individual antidepressants, even though they were classified as SSRIs and SNRIs. Similarly, such differences may also exist among individual congenital abnormalities even if they were categorized based on organ organization and function. Further population-based studies or animal research are needed to answer these questions.

Given the adverse effects of antidepressants on the health of offspring, we advocate that health providers consider non-pharmacological treatment in clinical practice to help women with depression during pregnancy. For example, randomized clinical trials have provided strong evidence that mindfulness-based cognitive therapy (MBCT) [73] and brief interpersonal therapy [74] can enhance psychological well-being in pregnant women. In addition, more than 20 states currently ban or restrict abortion procedures in early pregnancy, which has reduced the rate of women terminating pregnancies for medical reasons. A recent Stanford Medicine's study suggested that abortion bans will increase the number of newborns with serious heart defects [75]. Considering the poor prognosis and difficult life faced by newborns

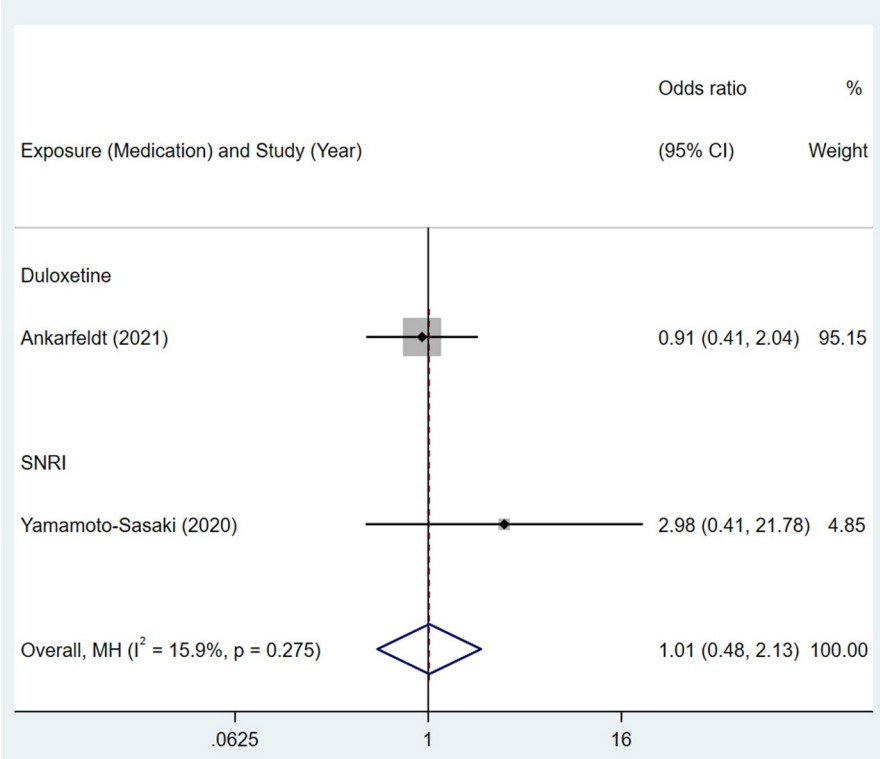

**Fig 27. Forest plot of the association between maternal SNRIs exposure and risk of congenital malformations of genital organs in offspring.**

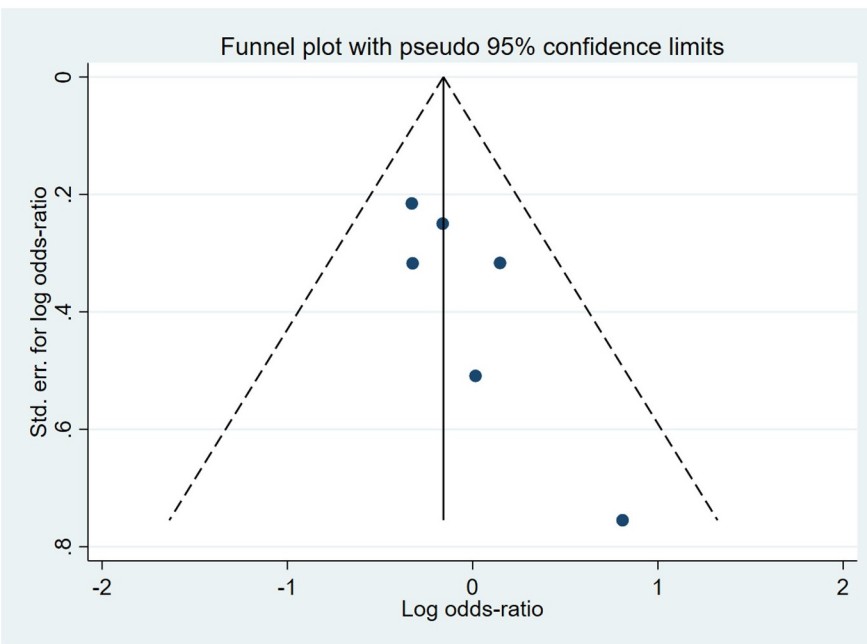

**Fig 28. Funnel plot of the association between maternal SSRIs exposure and risk of congenital malformations of genital organs in offspring.**

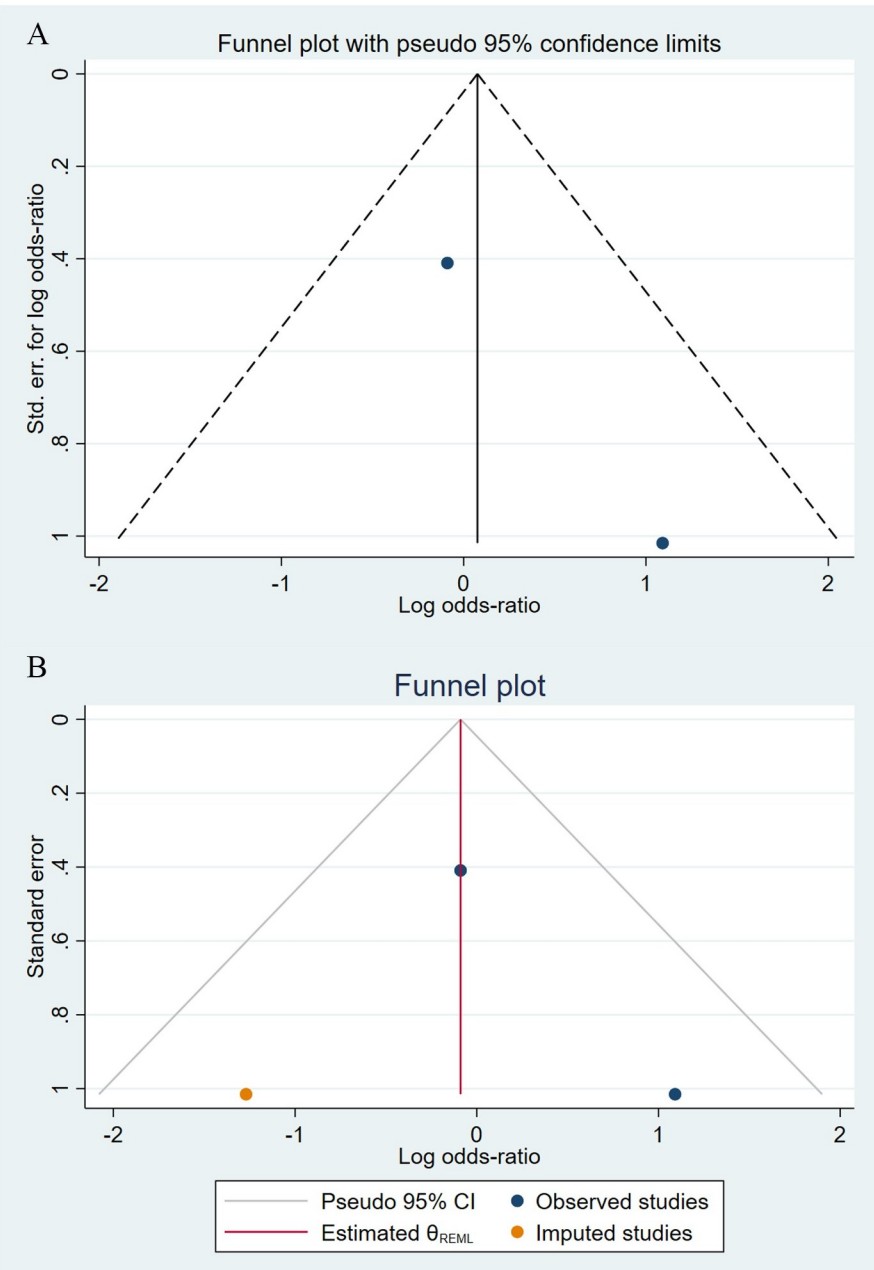

**Fig 29. Funnel plot of the association between maternal SNRIs exposure and risk of congenital malformations of genital organs in offspring.** A: Funnel plot before trim-and-fill methods. B: Funnel plot after trim-and-fill methods.

born with congenital malformations, prevention of congenital malformations is of great importance and non-pharmaceutical options should be considered when treating depression during pregnancy.

A major strength of the study was the large sample size and inclusion of multicenter studies from the United States, Europe, Canada, Japan, and Israel, which could provide sufficient statistical power to detect even small effects. However, there are a number of limitations. First, we were unable to conduct subgroup analyses on the timing of maternal

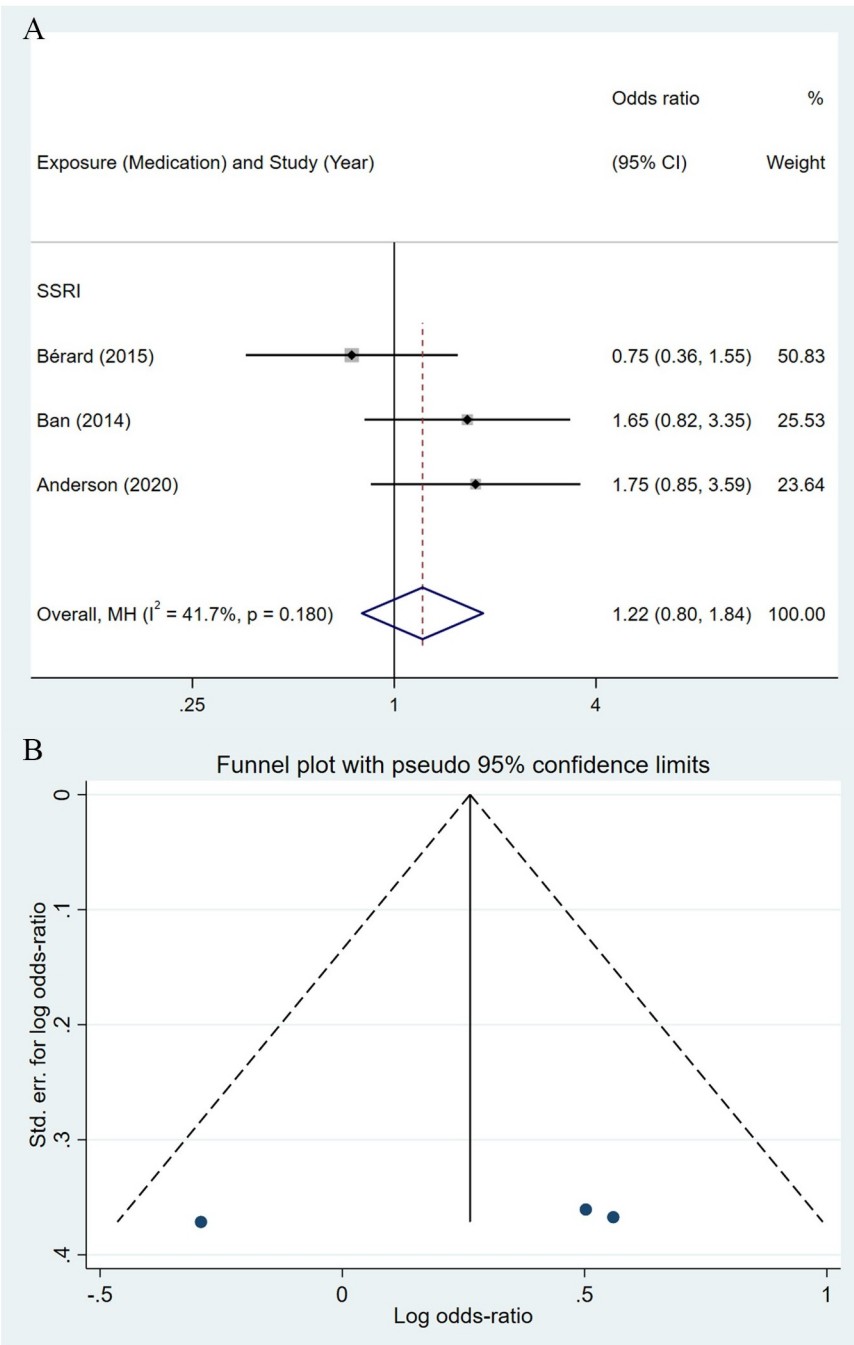

**Fig 30. Forest plot and funnel plot of the association between maternal SSRIs exposure and risk of congenital respiratory system malformations in offspring.** A: Forrest plot. B: Funnel plot.

antidepressants exposure (i.e. first, second, or third trimester) as a small number of studies were stratified by gestational age. However, previous studies found that maternal exposure to SSRIs or SNRIs during the first trimester of pregnancy had significant teratogenic effects on offspring [24, 56]. Second, besides the well-established effects of smoking [76], alcohol consumption [77], and maternal age [78] on congenital malformations, we were unable to adjust

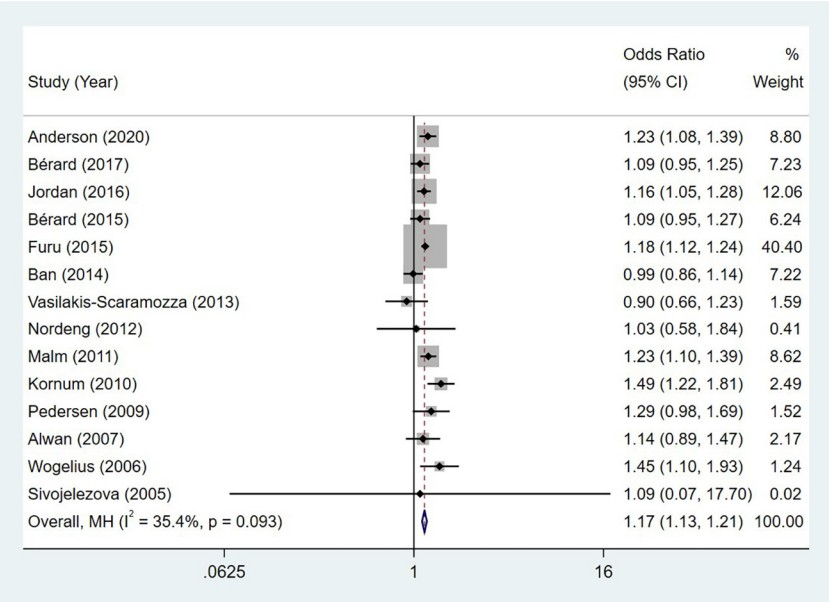

**Fig 31. Forest plot of the association between maternal SSRIs exposure and the overall malformation risk in offspring.**

the effects for these potential confounders as different sets of covariates were used in various studies and unavailability of individual-level data. Third, we used $I^2$ to identify heterogeneity and also estimated $\tau^2$ and its 95% CI to shed light on the between-study variability [20]. Most groups had estimated $\tau^2$ values zero or close to zero, and nearly all groups' 95% confidence

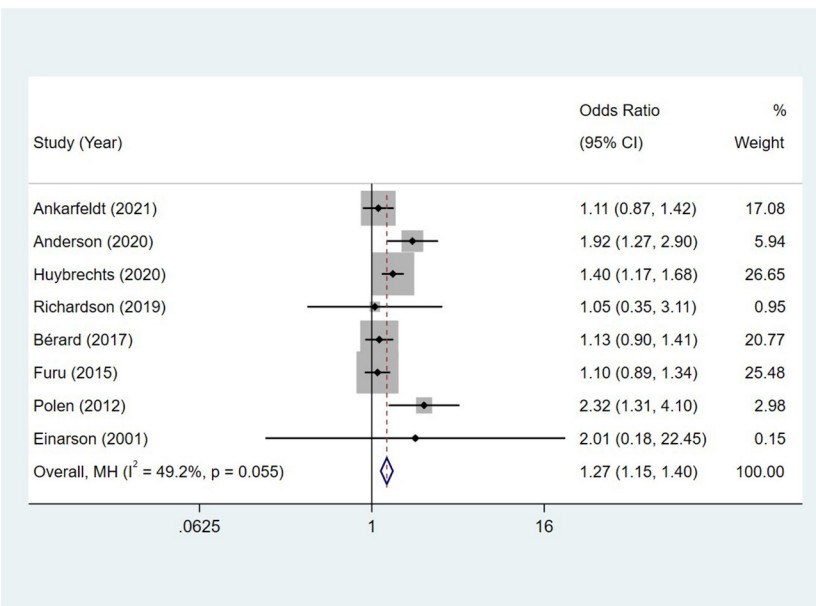

**Fig 32. Forest plot of the association between maternal SNRIs exposure and the overall malformation risk in offspring.**

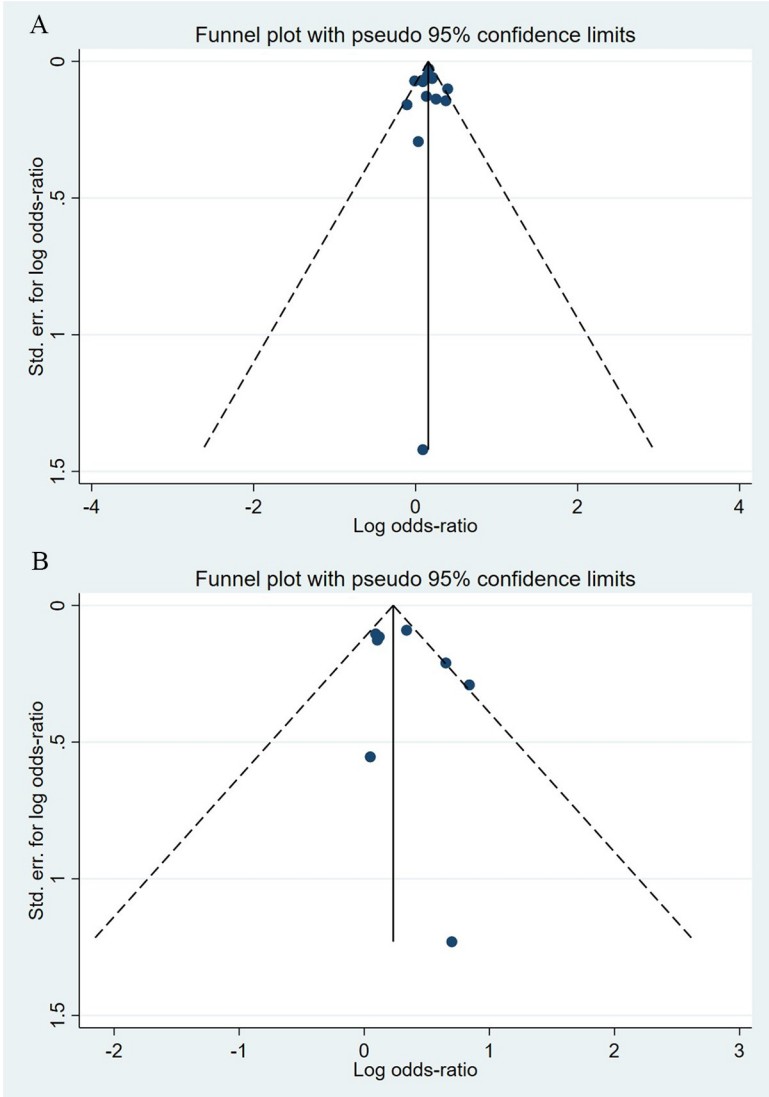

**Fig 33. Funnel plot of the association between maternal SSRIs/SNRIs exposure and the overall malformation risk in offspring.** A: Funnel plot for SSRIs. B: Funnel plot for SNRIs.

intervals included zero, see S4 Table, demonstrating the validity of the pooled analysis. Fourth, publication bias may affect research results. We detected publication bias for the association of maternal exposure to SNRIs with congenital cardiovascular abnormalities and anomalies of the kidney and urinary tract in offspring. However, when we used the trim and fill method to adjust for the potential publication bias, the recalculated pooled ORs just slightly decreased, and the conclusion remained unchanged. Aside from the publication bias, the asymmetry of the funnel plot may result from true heterogeneity, data irregularities, arte-facts or chance [22]. Even in the absence of publication bias, asymmetry is common in funnel plots when the number of studies is small [79]. Nevertheless, the majority of included studies were assessed to be of high quality, which may suggest a low risk of methodological bias and increase the reliability of the pooled results.

**Table 2. Odds ratios (ORs) and absolute risks for various subtypes of congenital malformations.**

| Malformation | SSRIs Pooled ORs | 95% CI | SNRIs Pooled ORs | 95% CI | Absolute risks * (Prevalence per 10,000 live births) | |
|---|---|---|---|---|---|---|
| Cardiovascular system | 1.25 | 1.20, 1.30 | 1.64 | 1.36, 1.97 | Tetralogy of Fallot | 4.61 |
| | | | | | Transposition of the great arteries | 3.71 |
| | | | | | Atrioventricular septal defect | 5.38 |
| | | | | | Coarctation of the aorta | 5.57 |
| | | | | | Pulmonary valve atresia and stenosis | 9.51 |
| | | | | | Tricuspid valve atresia and stenosis | 1.68 |
| | | | | | Single ventricle | 0.75 |
| | | | | | Hypoplastic left heart syndrome | 2.60 |
| | | | | | Double outlet right ventricle | 1.67 |
| | | | | | Common truncus (truncus arteriosus) | 0.64 |
| Kidney and urinary tract | 1.14 | 1.02, 1.27 | 1.63 | 1.21, 2.20 | - | |
| Nervous system | 1.07 | 0.93, 1.23 | 2.28 | 1.50, 3.45 | Anencephaly | 2.15 |
| | | | | | Encephalocele | 0.95 |
| | | | | | Spina bifida | 3.63 |
| Digestive system | 1.11 | 1.01, 1.21 | 2.05 | 1.60, 2.64 | Esophageal atresia/tracheoesophageal fistula | 2.41 |
| | | | | | Rectal and large intestinal atresia/stenosis | 4.46 |
| | | | | | Cleft lip with cleft palate | 6.40 |
| | | | | | Cleft lip without cleft palate | 3.56 |
| | | | | | Cleft palate | 5.93 |
| Abdominal birth defect | 1.33 | 1.16, 1.53 | 2.91 | 1.98, 4.28 | Diaphragmatic hernia | 2.78 |
| | | | | | Gastroschisis | 5.12 |
| | | | | | Omphalocele | 2.40 |
| Musculoskeletal system | 1.44 | 1.32, 1.56 | 0.90 | 0.60, 1.36 | Clubfoot | 16.86 |
| | | | | | Limb defects | 5.15 |
| Eye, ear, face and neck | 1.00 | 0.80, 1.26 | 1.44 | 0.82, 2.55 | Anophthalmia/microphthalmia | 1.91 |
| Genital organs | 0.84 | 0.65, 1.07 | 0.91 | 0.46, 1.84 | - | |
| Respiratory system | 1.22 | 0.80, 1.84 | 1.51 | 0.49, 4.70 | - | |

*Data & Statistics on Birth Defects is available from https://www.cdc.gov/ncbddd/birthdefects/data.html.

## Conclusion

These meta-analysis findings have important implications for clinicians who prescribe antide-pressant medications to their pregnant patients. Teratogenic effects of certain medications are well-documented, however clinicians must consider risk-benefit ratios and patient history when making prescribing decisions.

## Supporting information

**S1 Appendix. Literature search strategy.**
(DOCX)

**S1 Table. Quality assessment of cohort studies in the meta-analysis.**
(DOCX)

**S2 Table. Quality assessment of case-control studies in the meta-analysis.**
(DOCX)

**S3 Table. PRISMA for systematic reviews and meta-analyses checklist.**
(DOCX)

**S4 Table. Analysis of absolute between-study variability.**
(DOCX)

## Acknowledgments

We thank all members of the multidisciplinary research team for their support of this work.

## Author Contributions

**Conceptualization:** Weiyi Huang, Robin L. Page, Samiran Sinha.

**Data curation:** Weiyi Huang.

**Formal analysis:** Weiyi Huang, Samiran Sinha.

**Funding acquisition:** Robin L. Page, Theresa Morris, Susan Ayres, Alva O. Ferdinand, Samiran Sinha.

**Investigation:** Weiyi Huang.

**Methodology:** Weiyi Huang, Samiran Sinha.

**Supervision:** Robin L. Page, Samiran Sinha.

**Visualization:** Weiyi Huang.

**Writing – original draft:** Weiyi Huang.

**Writing – review & editing:** Weiyi Huang, Robin L. Page, Theresa Morris, Susan Ayres, Samiran Sinha.

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
