## [Decision Letter · Decision Letter 0]

13 Oct 2023

PONE-D-23-26655Maternal exposure to SSRIs or SNRIs and the risk of congenital abnormalities in offspring: A systematic review and meta-analysisPLOS ONE

Dear Dr. Huang,

Thank you for submitting your manuscript to PLOS ONE. After careful consideration, we feel that it has merit but does not fully meet PLOS ONE’s publication criteria as it currently stands. Therefore, we invite you to submit a revised version of the manuscript that addresses the points raised during the review process.

We look forward to receiving your revised manuscript.

Kind regards,

Carmen Concerto

Academic Editor

PLOS ONE

Journal Requirements:

"We acknowledge the support from Texas A&M Saving Babies X-Grant Project. "

"This work was funded by an internal grant from Texas A&M University Division of Research and was awarded to the following authors: RP, TM, SA, AF, SS. The Grant number is 290414-00001. The URL for the funder is https://vpr.tamu.edu/find-funding/. The funders had no role in study design, data collection and analysis, decision to publish, or preparation of the manuscript."

Reviewers' comments:

Reviewer's Responses to Questions

**Comments to the Author**

1. Is the manuscript technically sound, and do the data support the conclusions?

Reviewer #1: Yes

Reviewer #2: Yes

Reviewer #3: Yes

2. Has the statistical analysis been performed appropriately and rigorously? 

Reviewer #1: Yes

Reviewer #2: Yes

Reviewer #3: Yes

3. Have the authors made all data underlying the findings in their manuscript fully available?

Reviewer #1: Yes

Reviewer #2: Yes

Reviewer #3: Yes

4. Is the manuscript presented in an intelligible fashion and written in standard English?

Reviewer #1: Yes

Reviewer #2: Yes

Reviewer #3: Yes

5. Review Comments to the Author

Reviewer #1: 1. The paper appears to have a well-defined research question and design, conducting a meta-analysis to examine the association between maternal exposure to SSRIs and SNRIs during pregnancy and congenital malformations in offspring. Meta-analyses are valuable for synthesizing existing research.

2. The paper provides a detailed methodology, including data sources, search strategies, inclusion criteria, and statistical analysis methods. This transparency is crucial for assessing the rigor of the study.

3. The paper presents a comprehensive analysis of the results, including forest plots, pooled odds ratios, and confidence intervals for various congenital malformations. The findings are well-documented and reported clearly.

4. The Discussion section is thorough, providing a thoughtful interpretation of the results, their implications, and potential clinical relevance. The authors acknowledge limitations and suggest areas for future research.

5. The paper is generally well-written, with clear language and organization. It follows a standard structure for scientific research papers.

Reviewer #2: Paper PONE-D-23-26655 by Weiyi Huang

Maternal exposure to SSRIs or SNRIs and the risk of congenital abnormalities in offspring: A systematic review and meta-analysis

1. Overall opinion

Selective serotonin reuptake inhibitors (SSRIs) and serotonin and norepinephrine reuptake inhibitors (SNRIs) are the most common antidepressants prescribed during pregnancy, Depression is a relatively frequent disorder, and questions are raised on the risk of congenital anomalies related to these drugs. The paper intends to give an answer based on large numbers through meta-analyses of published papers. The paper is quite long, with many tables / figures, which has to be accepted by the journal, but it gives major information.

1. Recommendations

This paper will deserve publication after minor revision.

2. Detailed comments

a. Introduction

Introduction shows quite well the importance and the relevance of the raised question and is based on recent literature.

b. Material and methods

This section is well presented and is almost ok. The meta-analyses and statistical methodology is accurate. However, it would be necessary to specify in this section the categories of malformations looked for by authors. It would be also interesting to add, if possible, an analysis of the overall malformation rate. In effect, the high number of categories tested result in significant and non-significant results depending on anomalies categories and a global malformation rate would allow to better evaluate the real global risk. Moreover, it would also be interesting to give an estimate of the crude malformation percentages, which may allow the reader to see the real impact of the tested drugs on each malformation.

c. Results

Results are very well expressed and easy to read. As expressed above, it would also be interesting to give an estimate of the crude malformation risk

d. Discussion:

This section is well written, analysing the literature, the biological hypotheses, and the strengths and weaknesses. It would be interesting to have also a discussion on the points discussed by the reviewer above.

Reviewer #3: I had the pleasure of reviewing your manuscript, and I found your work very interesting and well-presented. However, I noticed that your analysis lacks the tau measure for assessing heterogeneity.

In your article, you only reported the I^2 measure. While this is an important measure, it is only a relative measure of heterogeneity and does not allow for assessing the prediction interval. The tau measure, on the other hand, can provide an estimate of absolute heterogeneity among the studies.

I suggest including the tau measure in your analysis. I believe that this addition could further enrich your work and provide readers with a more comprehensive understanding of the results.

Thank you for your attention, and I look forward to your revision.

6. PLOS authors have the option to publish the peer review history of their article (what does this mean?). If published, this will include your full peer review and any attached files.

Reviewer #1: No

Reviewer #2: **Yes: **Jacques de Mouzon, MD, MPH

Reviewer #3: **Yes: **Antonio Di Francesco

---

## [Author Response · Author response to Decision Letter 0]

8 Nov 2023

Response to Reviewer 1:

Thank you for recognizing the significance of our research and thank you so much for your positive comments.

Response to Reviewer 2:

Thank you for your comments and suggestions. We have specified the categories of malformations in the ‘Methods’ section and added an analysis of the association between the overall malformation and the drugs, SSRI and SNRI in the ‘Results’ section. The ‘Discussion’ section contains comments related to this analysis.

Regarding your comment on the crude malformation percentage, we cannot calculate that percentage because our analysis included both cohort and case-control studies. Although cohort data allow prevalence calculation, case-control data do not allow prevalence estimation. However, Table 2 contains the crude prevalence of the malformations; these numbers were extracted from the CDC webpage. Hope it is acceptable to you.

Response to Reviewer 3:

Thank you for your comments and suggestions. We have added the tau^2 measure to our analysis. Please see the ‘Statistical Analysis’ and ‘Discussion’ sections and the supplementary Table S4. In most cases, the value of tau^2 is equal to 0 or close to 0, indicating the studies are sufficiently similar or comparable to be pooled together.

---

## [Editor Report · Decision Letter 1]

14 Nov 2023

Maternal exposure to SSRIs or SNRIs and the risk of congenital abnormalities in offspring: A systematic review and meta-analysis

PONE-D-23-26655R1

Dear Dr. Huang,

We’re pleased to inform you that your manuscript has been judged scientifically suitable for publication and will be formally accepted for publication once it meets all outstanding technical requirements.

Kind regards,

Carmen Concerto

Academic Editor

PLOS ONE
---

## [Editor Report · Acceptance letter]

17 Nov 2023

PONE-D-23-26655R1 

Maternal exposure to SSRIs or SNRIs and the risk of congenital abnormalities in offspring: A systematic review and meta-analysis 

Dear Dr. Huang:

I'm pleased to inform you that your manuscript has been deemed suitable for publication in PLOS ONE. Congratulations! Your manuscript is now with our production department. 

Kind regards, 

on behalf of

Dr. Carmen Concerto 

Academic Editor

PLOS ONE